# DDSVM: A Differentiable Framework for Deep Support Vector Machines with Iterative Geometry-Aware Optimization

**Yirun Ding** [1]   **Zhihui Lai** [1]

## Abstract

Modern deep networks optimize features via cross-entropy without explicit margin maximization, while classical Support Vector Machines maximize the margin on a fixed feature space. Hybrid Deep-SVM approaches typically treat the deep feature extractor as a static input to an SVM head, so SVM geometry never flows back to shape representation learning. We propose the **Differentiable Deep Support Vector Machine (DDSVM)**, an iterative framework that alternates between re-fitting a linear SVM on $L_2$-normalized features and updating the backbone to pull each feature along the SVM-defined margin-widening direction, treating the decision boundary as a dynamic teacher for representation learning. Under spherical normalization the resulting feature gradient lies in the tangent space and optimization focuses exclusively on angular separability. DDSVM reduces top-1 error by **5.6–8.3%** on image classification benchmarks under data scarcity.

## 1. Introduction

Deep Neural Networks (DNNs) are the dominant approach to representation learning, achieving state-of-the-art performance across computer vision and natural language processing tasks (LeCun et al., 2015; Bengio et al., 2013; Vaswani et al., 2017; Devlin et al., 2019). Despite this empirical success, the traditional Support Vector Machine (SVM) remains a powerful tool in statistical learning theory, valued for its geometric interpretation and the principle of structural risk minimization via margin maximization (Cortes & Vapnik, 1995). Consequently, integrating the robust generalization capabilities of SVMs with the powerful feature extraction of DNNs has become a significant area of interest.

Early attempts to combine these methodologies, often termed *Deep SVMs*, typically replaced the standard Softmax layer with a linear SVM classifier or Cross-Entropy (CE) loss with Hinge loss (Tang, 2013; Niu & Suen, 2012). While these approaches successfully enforce margin constraints, they often treat the feature embedding as a static input for the classifier or rely on passive gradient updates that fail to fully exploit the geometric properties of the decision boundary. Recent theoretical studies have suggested an intrinsic connection between CE loss and hard-margin SVMs under certain conditions (Mao et al., 2023; Soudry et al., 2018), yet standard DNN optimization lacks an explicit mechanism to actively reshape the feature manifold based on the optimal separating hyperplane.

In this work, we propose a Differentiable Deep Support Vector Machine (DDSVM) framework that moves beyond static embeddings by introducing a dynamic, geometry-aware iterative optimization process.[1] Unlike traditional end-to-end training, our approach views the neural network as a trainable spherical mapping from data to a unit hypersphere, drawing inspiration from early deep face representation works (Taigman et al., 2014) and recent large-margin angular embeddings (Liu et al., 2017; Deng et al., 2019). We introduce a novel "Squeeze" mechanism that alternates between constructing the optimal SVM hyperplane and actively pushing feature points along the normal vector to maximize the geometric margin. The Squeeze mechanism lets the decision boundary act as a dynamic "teacher", guiding the representation learning process.

Furthermore, to address the challenge of hard samples near the decision boundary, we extend our framework with a Geometric Adversarial Squeeze (GAS) mechanism. Drawing inspiration from the Fast Gradient Sign Method (FGSM) (Goodfellow et al., 2015) and Virtual Adversarial Training (VAT) (Miyato et al., 2018), we reuse adversarial perturbation techniques to find the most vulnerable directions in the feature space. By minimizing the distance between these worst-case perturbations and ideal geometric targets, DDSVM produces a compact intra-class distribution and a widened inter-class margin.

---

[1]College of Computer Science and Software Engineering, Shenzhen University, 518060, Shenzhen, China. Correspondence to: Zhihui Lai <lai_zhi_hui@163.com>.

*Proceedings of the 43rd International Conference on Machine Learning*, Seoul, South Korea. PMLR 306, 2026. Copyright 2026 by the author(s).

---

[1]Source code is available at https://github.com/Four-Yi/DDSVM-ICML26.

The remainder of this paper is organized as follows. In Section 2, we discuss relevant works, including the fundamentals of SVMs, historical attempts to embed SVMs into neural networks, the theoretical links between CE loss and margin maximization, and the background of adversarial training methods like VAT. Section 3 introduces the DDSVM framework formally, detailing the iterative optimization pipeline, the spherical gradient derivations, and the GAS mechanism. Section 4 presents experimental results on image classification benchmarks, demonstrating the superiority of our geometry-driven approach over static baselines. Finally, Section 5 concludes the paper.

**Conflict of Interest Disclosure.** The authors declare no financial conflicts of interest. All authors are affiliated with Shenzhen University in an academic capacity, and no commercial entity commissioned, funded, or evaluated this work.

## 2. Related Work

In this section, we review the foundational works motivating our framework, ranging from the integration of SVMs with deep networks to the theoretical understanding of implicit regularization, and finally to adversarial strategies for manifold smoothing.

### 2.1. Deep Learning with SVMs

The Support Vector Machine is a standard tool in statistical learning, known for its max-margin principle which minimizes structural risk (Boser et al., 1992; Cortes & Vapnik, 1995). With the rise of Deep Learning, integrating the feature learning power of Convolutional Neural Networks (CNNs) with the generalization capability of SVMs became a natural pursuit.

Early hybrid approaches trained a stand-alone SVM head on fixed deep features extracted from a pre-trained CNN (Niu & Suen, 2012). Tang (2013) later moved beyond this two-stage setup by directly optimizing an $L_2$-SVM objective at the output layer with Hinge loss, demonstrating gains over Softmax on certain recognition tasks. Subsequent works extended this by introducing globally differentiable approximations of the Hinge loss or hybrid architectures for specific applications (Shuvo et al., 2020; Charabi et al., 2025). However, these methods mostly treat the feature embedding as a *static* input for the classifier during the margin maximization step, or rely on passive gradient updates.

### 2.2. Linking Cross-Entropy Loss to Hard-Margin SVMs

A fundamental question in deep learning is why overparameterized networks trained with Cross-Entropy (CE) loss generalize well even without explicit regularization.

Recent theoretical breakthroughs have linked this phenomenon to the max-margin principle. Mao et al. (2023) provided H-consistency bounds for comp-sum losses, including CE, suggesting an implicit alignment with margin maximization. While earlier studies by Soudry et al. (2018) rigorously proved that for linearly separable data, gradient descent on the unregularized logistic loss converges to the $L_2$ max-margin solution, our framework addresses the practical limitations of this implicit process.

A critical limitation identified by Soudry et al. (2018) is that this convergence is asymptotically **extremely slow**: logarithmic in the number of iterations ($O(1/\log t)$). This explains why training often needs to continue long after classification error reaches zero. Our work can be viewed as a response to this theoretical insight: instead of waiting for the slow, implicit convergence of gradient descent to find the max-margin direction, it explicitly and aggressively optimizes this geometric structure via a feature-space squeezing mechanism, effectively accelerating the margin maximization process.

### 2.3. Adversarial Perturbation and Manifold Smoothing

The concept of adversarial examples was first revealed by Szegedy et al. (Szegedy et al., 2014), who discovered that deep neural networks are vulnerable to imperceptible perturbations.

The **Fast Gradient Sign Method (FGSM)** was subsequently proposed by Goodfellow et al. (2015) to efficiently generate such adversarial examples by perturbing inputs in the direction of the cost function's gradient sign ($\epsilon \cdot \text{sign}(\nabla_x J)$). This approach uses the near-linear behaviour of deep networks in high dimensions to find worst-case perturbations under an $L_\infty$ norm constraint, and has become a foundational technique in adversarial machine learning and robust optimization (Akhtar & Mian, 2018; Yuan et al., 2019; Bai et al., 2021).

Extending this concept beyond supervised settings, **Virtual Adversarial Training (VAT)** (Miyato et al., 2018) utilizes the robustness of the conditional label distribution $p(y|x)$ around each input data point against local perturbations to regularize models. Unlike standard adversarial training that requires label information to compute adversarial directions, VAT defines a **virtual adversarial direction** by maximizing the divergence between the model's outputs on the original and perturbed inputs without relying on ground-truth labels, making it particularly effective for semi-supervised learning scenarios where unlabeled data is abundant.

While these methods primarily focus on the local smoothness of the probabilistic output distribution (e.g., minimizing KL-divergence), recent research suggests that adversarial-like perturbations can also be leveraged to shape the under-

lying geometric properties of the feature space (Li et al., 2019; Wen et al., 2016). Our work builds upon this lineage but shifts the focus from distribution smoothing to explicit geometric margin expansion. By adapting gradient-based perturbations into a supervised context, we aim to enforce a safety margin in the embedding space, mapping worst-case feature-space perturbations back to ideal geometric targets defined by the SVM boundary.

## 3. Methodology

In this section, we formally present the Differentiable Deep Support Vector Machine (DDSVM). We first frame the neural network as a trainable spherical mapping within the SVM objective. We then introduce the overall loss function, followed by a detailed derivation of the geometry-aware target synthesis, the gradient alignment property on the hypersphere, and the evolution from consistency regularization to Geometric Adversarial Squeeze (GAS).

A summary of the notation used throughout this section is provided in Appendix A.1.

### 3.1. The Neural Network as a Trainable Kernel

Traditionally, SVMs often rely on a pre-defined kernel function $K(\mathbf{x}_i, \mathbf{x}_j) = \langle \phi(\mathbf{x}_i), \phi(\mathbf{x}_j) \rangle$ to map input data into a high-dimensional feature space $\mathcal{H}$ (Boser et al., 1992; Cortes & Vapnik, 1995; Schölkopf & Smola, 2002), where the mapping $\phi(\cdot)$ is static. In contrast, DDSVM treats the deep neural network as a **trainable spherical mapping** parameterized by $\Theta$.

Let $F(\mathbf{x}; \Theta) : \mathcal{X} \to \mathbb{R}^d$ denote the raw feature output of the backbone network. To eliminate scale ambiguity and align with the geometric interpretation of cosine margins, we impose a spherical constraint via $L_2$ normalization. The effective mapping is defined as:

$$\phi(\mathbf{x}; \Theta) = \hat{\mathbf{f}} = \frac{F(\mathbf{x}; \Theta)}{\|F(\mathbf{x}; \Theta)\|_2} \in \mathbb{S}^{d-1} \qquad (1)$$

where $\mathbb{S}^{d-1}$ represents the unit hypersphere.

The optimization goal of DDSVM is to jointly learn the parameters $\Theta$ and the linear decision hyperplanes $\{\mathbf{w}_c, b_c\}_{c=1}^K$ such that the geometric margin is maximized on the hypersphere. While the hinge objective is differentiable almost everywhere and can be optimized end-to-end, joint optimization of the kernel mapping and the decision hyperplane mixes the convex SVM problem with the non-convex backbone optimization and produces a moving geometric teacher. We therefore propose an iterative alternating optimization strategy in which the SVM boundary is solved to optimality and frozen at each refinement step. The design draws on early CNN-SVM combinations (Niu & Suen, 2012; Tang, 2013) and on the alternating multi-module training scheme

of SADH (Shen et al., 2018), which interleaves deep representation learning with a separately-optimized discrete objective.

The framework shown in Figure 1 comprises two phases. Phase I is a standard cross-entropy warm-up that drives the backbone to a linearly-separable feature space; the transition to Phase II is governed by a simple convergence trigger based on training accuracy (Section 3.3). Phase II is the refinement loop in which the geometry-aware optimization actually takes place: at the beginning of each epoch the linear SVMs are re-fit on the current frozen features, and the backbone is then updated to pull each feature toward an SVM-defined geometric target. These two operations alternate for as many refinement epochs as desired. The number of epochs is a hyperparameter rather than a design constraint, and after Phase II the cycle could in principle be re-entered to extract new features and retrain the SVMs. The total loss function $\mathcal{L}_{total}$ governing the representation update (Phase II) is defined as the Mean Squared Error (MSE) between the active feature projection and a synthesized geometric target $\mathbf{T}$:

$$\mathcal{L}_{total}(\Theta) = \frac{1}{2N} \sum_{i=1}^{N} \|\hat{\mathbf{f}}_i(\Theta)_{active} - \mathbf{T}_i\|_2^2 \qquad (2)$$

Here, $\hat{\mathbf{f}}_i(\Theta)_{active}$ represents the feature vector potentially augmented by perturbations (discussed in Section 3.5.2), and $\mathbf{T}_i$ is the static geometric target derived from the SVM boundary.

### 3.2. Geometry-Aware Target Synthesis

To actively maximize the margin, we synthesize an Ideal Target $\mathbf{T}_i$ for each sample $\mathbf{x}_i$ based on its current geometric relation to the decision boundary.

#### 3.2.1. MARGIN STATISTICS & DYNAMIC CALIBRATION

At the beginning of each optimization epoch, we calculate the signed distance of sample $i$ (belonging to class $y_i$) to its corresponding hyperplane defined by normal vector $\mathbf{w}_{y_i}$ (where $\|\mathbf{w}_{y_i}\|_2 = 1$):

$$dist_i = \mathbf{w}_{y_i}^T \hat{\mathbf{f}}_i + b_{y_i} \qquad (3)$$

We then compute the class-wise dynamic bounds:

$$D_{max,c} = \max_{j \in y_j = c} (dist_j)$$

$$D_{min,c} = \min_{j \in y_j = c} (dist_j)$$

These bounds define the current "geometric spread" of the class.

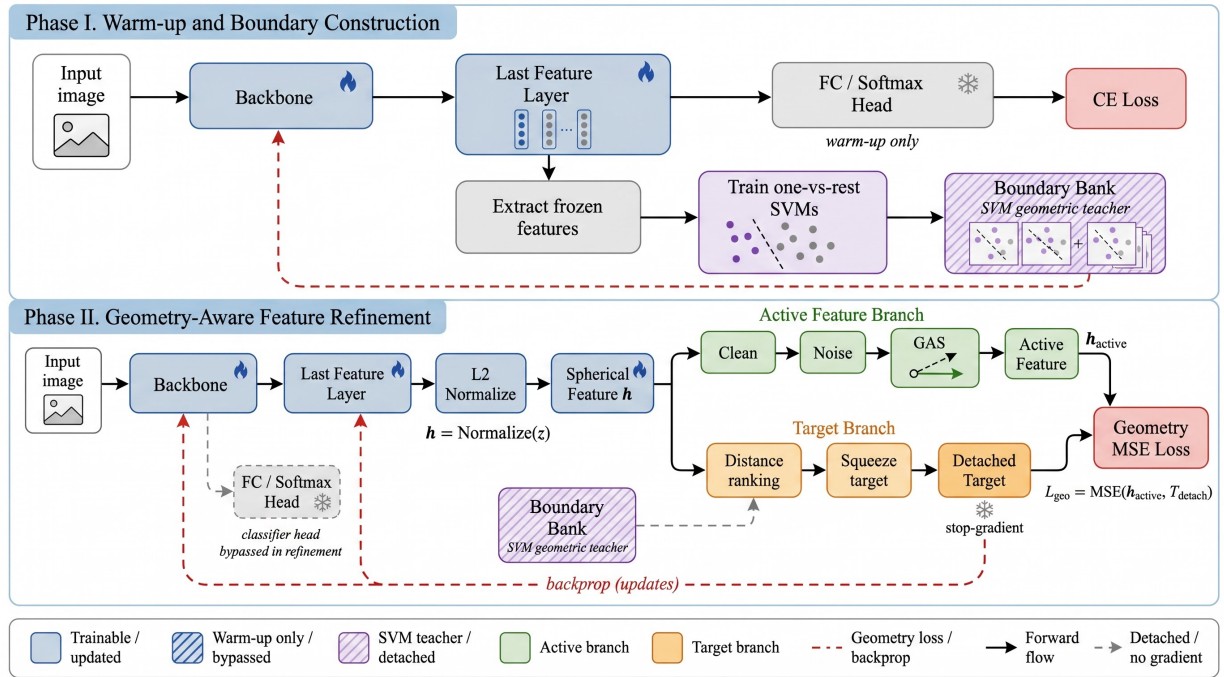

*Figure 1.* Architectural overview of the proposed DDSVM framework. **Phase I (Warm-up and Boundary Construction):** the backbone is trained with cross-entropy on the FC/Softmax head; once $\text{Acc}_{train} \geq \tau$ (Section 3.3), the frozen features are passed to a Boundary Bank of linear SVMs that provide a fixed geometric teacher for Phase II. **Phase II (Geometry-Aware Feature Refinement):** the FC/Softmax head is bypassed; features are $L_2$-normalized to the unit hypersphere, optionally perturbed by GAS into a synthetic hard example, and pulled by an MSE loss towards a detached, ranking-based squeezed target derived from the Boundary Bank. The dashed arrows mark stop-gradient edges and the dotted line marks the bypassed classification head.

### 3.2.2. THE SQUEEZE MECHANISM

We propose a non-linear squeeze function that applies adaptive pressure. The normalized distance ratio $\rho_i$ is calculated as:

$$\rho_i = \frac{D_{max,y_i} - dist_i}{D_{max,y_i} - D_{min,y_i}} \quad (4)$$

A sample closer to the boundary ($dist_i \rightarrow D_{min}$) yields $\rho_i \rightarrow 1$, indicating high urgency. The squeeze magnitude $m_i$ is defined as:

$$m_i = \theta \cdot \text{sign}(\rho_i)|\rho_i|^p \cdot \left(D_{min,y_i} - \frac{D_{max,y_i}}{\gamma}\right) \quad (5)$$

where $\theta$ is the squeeze strength, $p$ is the squeeze power controlling non-linearity, and $\gamma$ scales the target offset from the positive-margin bound $D_{max,y_i}$. The exponent $p$ controls how broadly the push is distributed: at the two endpoints ($\rho = 0$ and $\rho = 1$) the push magnitude is independent of $p$, while for intermediate samples a smaller $p$ delivers a larger push and a larger $p$ leaves the interior nearly fixed (Figure 2).

### 3.2.3. REST CLASS REPULSION

To prevent feature collapse and expand the inter-class margin, we incorporate a repulsion force. The effective squeeze direction $\mathbf{d}_i$ is a composite vector:

$$\mathbf{d}_i = \text{Norm}\left(\mathbf{w}_{y_i} - \beta \cdot \frac{1}{K-1}\sum_{k \neq y_i}\mathbf{w}_k\right) \quad (6)$$

where $\beta$ controls the repulsion strength from other class centroids. Finally, the **Ideal Target** $\mathbf{T}_i$ is synthesized in the tangent space and detached from the computation graph:

$$\mathbf{T}_i = \text{stop\_gradient}\left(\hat{\mathbf{f}}_i - m_i \cdot \mathbf{d}_i\right) \quad (7)$$

### 3.3. Convergence Trigger between Phase I and Phase II

The switch from boundary construction (Phase I) to geometry-aware refinement (Phase II) is controlled by a simple accuracy-based criterion rather than a gradient-norm or margin-stability metric. Let $\text{Acc}_{train}(t)$ denote the training accuracy at epoch $t$. Phase II is entered at the first epoch satisfying

$$\text{Acc}_{train}(t) \geq \tau. \quad (8)$$

This threshold ensures that the SVMs in Phase II are fitted on features that already linearly separate the training set, which lets LibLinear converge in seconds and makes the geometric targets informative; we use $\tau = 0.95$ as a representative

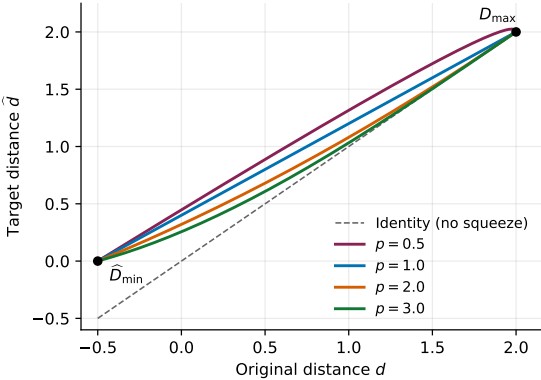

*Figure 2.* The squeeze function: target distance $\hat{d}$ as a function of the original distance $d$ for several values of the squeeze power $p$. All curves are anchored at $D_{\max}$ on the right and at the post-squeeze minimum $\hat{D}_{\min}$ on the left. The exponent $p$ controls how broadly the push is distributed across the interior: $p < 1$ pushes the interior upward more uniformly, $p = 1$ is a linear interpolation between the two anchors, and $p > 1$ restricts the push to samples close to the boundary while leaving the rest essentially fixed.

value. Running Phase II directly on a randomly initialized or severely underfit backbone is not viable. Exiting Phase I at this threshold also terminates the slow $O(1/\log t)$ implicit-margin convergence of cross-entropy (Soudry et al., 2018) and keeps the total training cost within the standard CE budget (see Section 4.5).

### 3.4. Spherical Constraint and Gradient Alignment

To ensure stable convergence in high-dimensional feature spaces, effective normalization strategies are essential. Unlike standard softmax classification which operates on unconstrained logits to match one-hot distributions, our framework directly optimizes the geometric arrangement of feature vectors. Following the classic metric learning paradigms established by methods like FaceNet (Schroff et al., 2015), SphereFace (Liu et al., 2017), CosFace (Wang et al., 2018) and ArcFace (Deng et al., 2019), we adopt a hard spherical constraint $\phi(\mathbf{x}) = \frac{F(\mathbf{x})}{\|F(\mathbf{x})\|_2}$, projecting all embeddings onto a unit hypersphere.

This constraint matters because layer-wise weight scaling in deep networks can approximately inflate feature norms while leaving angular directions largely unchanged (Liu et al., 2016). In the absence of such constraints, the network often minimizes loss simply by inflating feature magnitudes rather than improving angular separability, leading to a "feature collapse" where directions remain ambiguous. Enforcing the spherical constraint forces optimization onto the angular component. Inspired by BYOL's collapse-avoidance analysis (Grill et al., 2020), Appendix A.3 gives a post-hoc account of the radial shortcut this constraint closes.

*Remark* 3.1 (Gradient Orthogonality under $L_2$ Normaliza-

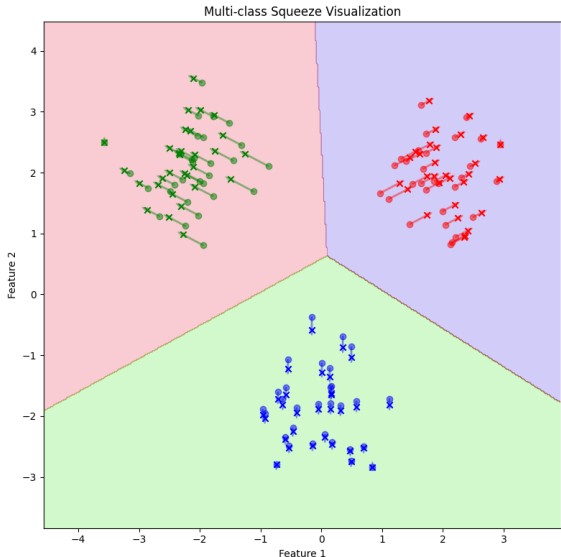

*Figure 3.* Visualization of the geometry-aware target synthesis in a multi-class setting. The arrows indicate the composite vector $\mathbf{d}_i$, which combines the intra-class attraction (towards correct normal) and inter-class repulsion (away from other class).

tion). Under $L_2$ normalization, the gradient backpropagated to the raw feature vector is strictly orthogonal to the feature vector itself, so any optimization step purely updates the angular component. We recall this well-known property of optimization on the unit hypersphere $\mathbb{S}^{d-1}$ to establish the basis for the subsequent design: backpropagation on $\mathbb{S}^{d-1}$ is naturally confined to the tangent space, which complements the radial pushing direction of the squeeze mechanism. See Appendix A.2 for the formal derivation.

With the feature space constrained to the hypersphere, we synthesize the geometric target $\mathbf{T}_i$ and detach it from the computation graph (stop-gradient), so that the squeeze function does not interfere with the gradient calculation. The SVM-defined boundary acts as a dynamic "Teacher", providing stable geometric anchors that the "Student" backbone approximates via simple MSE regression.

### 3.5. From Noise Consistency to Geometric Adversarial Squeeze

To further smooth the decision manifold and handle hard samples located in ambiguous regions, we extend the squeeze mechanism to a stochastic setting.

#### 3.5.1. CONSISTENCY REGULARIZATION

We first consider a local neighborhood around $\mathbf{f}_i$ by adding Gaussian noise $\mathbf{n} \sim \mathcal{N}(0, \sigma^2\mathbf{I})$. The consistency requirement enforces that the perturbed feature $\text{Norm}(\mathbf{f}_i + \mathbf{n})$ must also map to the same target $\mathbf{T}_i$. While effective for

general smoothness, isotropic noise is inefficient in high-dimensional spaces as it treats all directions equally.

### 3.5.2. GEOMETRIC ADVERSARIAL SQUEEZE (GAS)

In the context of SVM optimization, directions orthogonal to the decision boundary are critical. Standard random noise is inefficient in high-dimensional spaces as it rarely aligns with these fragile directions. To address this, we introduce GAS to explicitly mine the "most vulnerable" direction.

We define the adversarial perturbation $\delta_{adv}$ by maximizing the drop in the SVM score $S(\hat{\mathbf{f}}) = \mathbf{w}_{y_i}^T \hat{\mathbf{f}} + b_{y_i}$. To efficiently traverse the high-dimensional feature space, we adopt an $L_\infty$-norm constraint similar to FGSM:

$$\delta_{adv} = \epsilon_{adv} \cdot \text{sign}\left(-\nabla_{\hat{\mathbf{f}}} S(\hat{\mathbf{f}})\right) = -\epsilon_{adv} \cdot \text{sign}(\mathbf{w}_{y_i}) \quad (9)$$

By applying this perturbation, we generate a synthetic hard feature $\hat{\mathbf{f}}_{adv} = \text{Norm}(\mathbf{f}_i + \delta_{adv})$. The network is then updated to minimize the distance between this adversarial feature and the safe geometric target $\mathbf{T}_i$:

$$\mathcal{L}_{GAS} = \frac{1}{2}\|\hat{\mathbf{f}}_{adv} - \mathbf{T}_i\|_2^2 \quad (10)$$

Note the scaling effect of the sign operation. For a feature dimension $D$, the Euclidean magnitude of the perturbation is $\|\delta_{adv}\|_2 = \epsilon_{adv}\sqrt{D}$. In our experiments (where $D = 512$), a setting of $\epsilon_{adv} = 1.0$ yields a Euclidean displacement of $\approx 22.6$, transforming $\hat{\mathbf{f}}_{adv}$ from a locally jittered sample into a synthetic hard negative located deep within the error region (full scaling analysis in Appendix A.4). Optimizing $\mathcal{L}_{GAS}$ forces the network to collapse a wide cone of the feature space towards the class center, expanding the geometric margin.

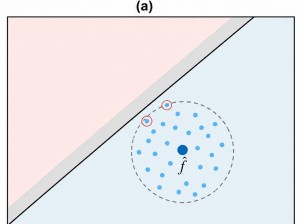 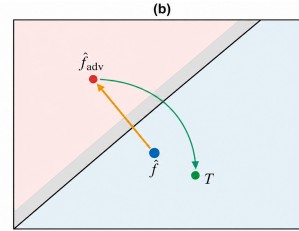

*Figure 4.* The Geometric Adversarial Squeeze (GAS) mechanism. **(a)** Isotropic Gaussian noise $\mathcal{N}(0, \sigma^2\mathbf{I})$ around $\hat{\mathbf{f}}$ samples all directions uniformly. **(b)** GAS generates a single deterministic perturbation $\delta_{adv} = \epsilon_{adv} \cdot \text{sign}(-\nabla_{\hat{\mathbf{f}}} S)$, orthogonal to the SVM boundary, producing the worst-case adversarial feature $\hat{\mathbf{f}}_{adv}$; the squeeze loss then pulls it back to the safe geometric target $\mathbf{T}$.

## 4. Experiments

In this section, we empirically validate the proposed DDSVM framework. Our evaluation is organized around two questions: (i) does DDSVM provide consistent gains over cross-entropy (CE) training across datasets and domains, especially in low-data regimes? (ii) how does it compare to angular-margin softmax losses and standard data-augmentation regularizers?

### 4.1. Experimental Setup

#### 4.1.1. DATASETS

We evaluate on four image classification benchmarks spanning two visual domains and two input resolutions: **CIFAR-10** and **CIFAR-100** (Krizhevsky et al., 2009) (natural images, $32\times32$); **STL-10** (Coates et al., 2011) (natural images, $96\times96$); and **SVHN** (Netzer et al., 2011) (street-view digits, $32 \times 32$). For each dataset we evaluate two regimes:

- **Full-data:** the standard training split.

- **Reduced (low-data):** a class-balanced subset comprising approximately $8\%$ of the training images (e.g. $4,000$ images for CIFAR-10 and STL-10) to simulate data scarcity.

To assess robustness against distribution shift, we adopt the standard **CIFAR-10-C** benchmark (Hendrycks & Dietterich, 2019), which applies 15 corruption types at 5 severity levels to the CIFAR-10 test set. We report both per-corruption error at severity 5 and the mean Corruption Error (mCE).

Standard data augmentations (random $32\times32$ crop with padding 4, horizontal flip) are applied to all CIFAR/STL-10 training images. SVHN follows its standard preprocessing protocol.

#### 4.1.2. ARCHITECTURES AND BASELINES

The main experiments use **ResNet-18** (He et al., 2016) adapted for CIFAR (initial max-pooling layer removed). All multi-seed numbers in the main paper are reported on this backbone. To check that the gains carry over to other architectures, we additionally report single-seed results on **ResNet-34** (He et al., 2016) and **WideResNet-28-10 (WRN-28-10)** (Zagoruyko & Komodakis, 2016) in Appendix A.7. The CE baseline uses a standard fully-connected classifier; for DDSVM, the FC layer is detached during Phase II, and the backbone is optimized against SVM-synthesized targets.

To contextualize DDSVM against geometry-aware losses, we additionally compare to three angular-margin softmax variants: **ArcFace** (Deng et al., 2019) ($s = 16, m = 0.2$), **CosFace** (Wang et al., 2018) ($s = 30, m = 0.35$), and **Large-Margin Softmax (LMS)** (Liu et al., 2016) ($s =$

*Table 1.* Top-1 error rates (%) across four datasets and two data regimes, ResNet-18, mean $\pm$ std over 3 seeds. "Full" uses the standard training split; "Reduced" uses $\sim 8\%$ of the training data. Lower is better.

| Regime | Method | CIFAR-10 | CIFAR-100 | STL-10 | SVHN |
|--------|--------|----------|-----------|--------|------|
| Full | CE | **4.64 $\pm$ 0.08** | **21.67 $\pm$ 0.19** | 19.45 $\pm$ 0.47 | **2.85 $\pm$ 0.08** |
|  | DDSVM | 4.70 $\pm$ 0.03 | 22.06 $\pm$ 0.45 | **17.43 $\pm$ 0.23** | 2.88 $\pm$ 0.01 |
| Reduced ($\sim$8%) | CE | 30.75 $\pm$ 5.27 | 68.25 $\pm$ 1.25 | 47.29 $\pm$ 3.75 | **6.08 $\pm$ 0.33** |
|  | DDSVM | **23.21 $\pm$ 0.77** | **62.67 $\pm$ 0.50** | **38.98 $\pm$ 2.09** | 6.79 $\pm$ 0.89 |

*Table 2.* Top-1 error (%) on CIFAR-10-C at severity 5 (ResNet-18, single seed for the corruption sweep). $\Delta$ = DDSVM $-$ CE; negative values (DDSVM better) are in **bold**.

| Corruption | CE | DDSVM | $\Delta$ | Corruption | CE | DDSVM | $\Delta$ |
|-----------|-----|-------|----------|-----------|-----|-------|----------|
| gaussian_noise | 75.05 | 75.50 | +0.45 | snow | 21.99 | 22.23 | +0.24 |
| shot_noise | 67.93 | 68.78 | +0.85 | frost | 35.17 | 35.23 | +0.06 |
| impulse_noise | 75.31 | 75.00 | $-$0.31 | fog | 26.51 | 25.12 | $-$1.39 |
| defocus_blur | 44.56 | 42.91 | $-$1.65 | brightness | 8.34 | 8.94 | +0.60 |
| glass_blur | 48.05 | 46.80 | $-$1.25 | contrast | 65.50 | 64.78 | $-$0.72 |
| motion_blur | 31.80 | 31.51 | $-$0.29 | elastic_transform | 21.69 | 21.90 | +0.21 |
| zoom_blur | 37.54 | 33.90 | $-$3.64 | pixelate | 48.62 | 49.64 | +1.02 |
|  |  |  |  | jpeg_compression | 25.63 | 24.58 | $-$1.05 |

**mCE @ severity 5**: CE 42.25%, DDSVM **41.79**% ($-$0.46).

$16, m = 0.2$). We also compare to two widely used data-augmentation regularizers, **Mixup** (Zhang et al., 2018) and **CutMix** (Yun et al., 2019), both stand-alone and combined with DDSVM.

### 4.1.3. IMPLEMENTATION DETAILS

Models are trained with **SGD** (momentum 0.9, weight decay $5 \times 10^{-4}$). Phase I uses a **cosine annealing** (Loshchilov & Hutter, 2017) learning-rate schedule, and Phase II is initialized with a small learning rate ($10^{-3}$) decayed by cosine annealing over 10 additional epochs. The phase switch follows Eq. (8): Phase II is entered as soon as $\text{Acc}_{train} \geq \tau$. The linear SVMs in Phase I are solved with LibLinear via Scikit-learn (Pedregosa et al., 2011) and **re-fit at every epoch** during Phase II. Default hyperparameters are $\theta = 1.0$, $p = 1.0$, $\beta = 0.1$, $\gamma = 1.0$, SVM regularization $C = 0.01$, and GAS perturbation strength $\epsilon_{adv} = 1.0$; these are kept fixed across all datasets unless stated otherwise. Detailed dataset configurations, baseline implementations, and hardware specifications are in Appendices A.8, A.12 and A.11.

### 4.2. Main Results across Datasets and Regimes

Table 1 summarizes results across all four datasets and both regimes. Three observations stand out.

**Consistent gains in the low-data regime on natural images.** On all three natural-image datasets, DDSVM reduces low-data error substantially: $-7.54\%$ on CIFAR-10, $-5.58\%$ on CIFAR-100 and $-8.31\%$ on STL-10. In addition, the variance across seeds is reduced by up to $7\times$ (CIFAR-10: $5.27 \to 0.77$), indicating that the SVM-driven

geometric target stabilizes optimization when the training set under-determines the manifold. On STL-10 the effect also extends to the full-data setting ($-2.02\%$), where the smaller training set per class (500 images) leaves room for the explicit margin.

**Diminishing returns in data-abundant settings.** In the full-data CIFAR-10/100 and SVHN cells DDSVM is statistically indistinguishable from CE ($+0.03$ to $+0.39\%$). This is consistent with the theoretical result of Soudry et al. (2018): when the data is abundant enough that gradient descent on CE has already implicitly approached the max-margin solution, explicit geometric refinement provides only marginal additional benefit. DDSVM is designed as a geometric regularizer for the data-scarce regime, not as a universal replacement for CE.

On SVHN, both regimes show only minor differences ($+0.03\%$ full, $+0.71\%$ reduced). SVHN's digit features are simple and largely class-aligned, leaving little geometric structure for SVM-guided refinement to reshape; we do not draw broader conclusions from this single domain.

### 4.3. Comparison with Angular-Margin and Augmentation Baselines

Table 3 compares DDSVM against angular-margin losses and augmentation regularizers under identical backbone and training budget. DDSVM (23.21%) outperforms ArcFace, CosFace and LMS (all in the 25.4–25.9% band) and also beats Mixup and CutMix used in isolation. Margin softmax variants enforce a static angular margin via a single loss term; DDSVM, in contrast, derives its target dynamically

*Table 3.* Top-1 error (%) on Reduced CIFAR-10 (8%, ResNet-18, 3 seeds). DDSVM is compared against angular-margin softmax losses and augmentation-based regularizers.

| Method | Error Rate |
|---|---|
| CE (baseline) | $30.75 \pm 5.27$ |
| ArcFace ($s = 16, m = 0.2$) | $25.92 \pm 2.82$ |
| CosFace ($s = 30, m = 0.35$) | $25.67 \pm 2.76$ |
| Large-Margin Softmax ($s = 16, m = 0.2$) | $25.38 \pm 0.82$ |
| CE + Label Smoothing | $27.29 \pm 2.47$ |
| CE + Mixup | $25.08 \pm 1.39$ |
| CE + CutMix | $25.17 \pm 1.03$ |
| **DDSVM (ours)** | $23.21 \pm 0.77$ |
| **DDSVM + Mixup** | **$17.67 \pm 1.22$** |
| **DDSVM + CutMix** | $17.88 \pm 0.18$ |

*Table 4.* Wall-clock training time (seconds, ResNet-18, RTX 4090 GPU + Ryzen 9 9950X CPU). DDSVM solves all SVMs on CPU via LibLinear on 512-d features. Phase 2 here includes the SVM time.

| Dataset | CE | DDSVM | Phase 2 | SVM | $\Delta$ |
|---|---|---|---|---|---|
| CIFAR-10 | 2874 | 1993 | 144 | 58 | $-31\%$ |
| CIFAR-100 | 2898 | 2802 | 137 | 418 | $-3\%$ |

from the up-to-date SVM boundary, which adapts to the current geometry of the training features and assigns larger pushing force to harder samples (the $|\rho_i|^p$ term in Eq. (5)).

Because DDSVM's geometric regularization and image-space augmentation address orthogonal sources of generalization error, the two combine cleanly: DDSVM + Mixup achieves $17.67\%$, a further $-5.54\%$ over DDSVM alone and $-7.41\%$ below the best stand-alone augmentation baseline.

### 4.4. Robustness on CIFAR-10-C

We evaluate robustness on the standard CIFAR-10-C benchmark (Table 2). DDSVM improves mCE at severity 5 from $42.25\%$ to $41.79\%$. Figure 5 visualizes the per-corruption gap $\Delta = \text{DDSVM} - \text{CE}$ sorted by sign and magnitude: DDSVM improves 8 of 15 corruptions, with the largest gains on geometric / spatial distortions (zoom_blur $-3.64$, defocus_blur $-1.65$, glass_blur $-1.25$, fog $-1.39$) and on the compression-style jpeg ($-1.05$). Small regressions concentrate on pixel-level noise (gaussian, shot, brightness, pixelate), with a worst-case degradation of $+1.02$. This pattern is consistent with the method's design: a wider SVM-defined angular margin protects against perturbations that shift the feature along smooth, geometrically structured directions, but offers little against high-frequency pixel noise that scrambles the angular code itself. Full per-severity tables are provided in Appendix A.9.

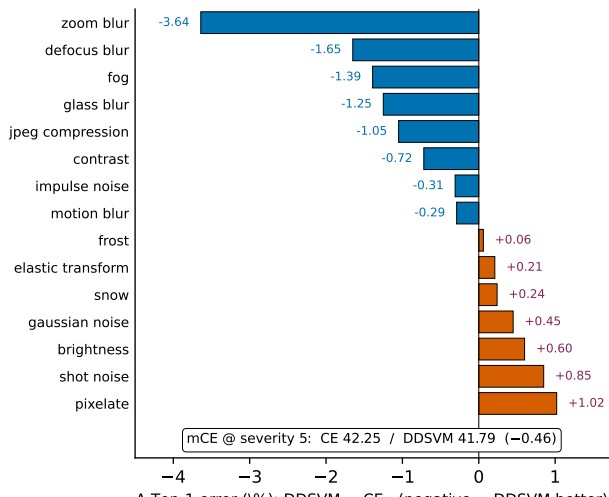

*Figure 5.* Per-corruption gap on CIFAR-10-C at severity 5, sorted by $\Delta = \text{DDSVM} - \text{CE}$ (lower is better for DDSVM). Blue bars: DDSVM improves over CE; orange bars: CE improves over DDSVM. Gains concentrate on geometric / spatial corruptions (blurs, fog) and on JPEG compression; small regressions concentrate on pixel-level noise.

### 4.5. Computational Efficiency

Repeatedly re-fitting linear SVMs during Phase II is a natural source of overhead. Table 4 and Figure 6 quantify it. Under the standard 200-epoch cosine schedule for CE, the full DDSVM pipeline completes within the same wall-clock budget. The pipeline consists of Phase I exiting at the accuracy threshold $\tau$ (with $\tau = 0.95$, this occurs at epoch $\approx 122$ for CIFAR-10 and $\approx 154$ for CIFAR-100 in our runs) plus a 10-epoch Phase II that includes all SVM fits. The SVM fits themselves account for only $2\%$ of total training time on CIFAR-10 and $15\%$ on CIFAR-100. We do not claim that DDSVM is intrinsically faster than CE: an early-stopped CE baseline could also reduce its budget. Rather, DDSVM provides a stopping criterion (the linearly-separable accuracy threshold $\tau$) together with a fixed refinement window, and matches or improves CE accuracy under this combined budget, so the SVM-fitting overhead does not translate into a wall-clock penalty in practice.

### 4.6. Ablation: Squeeze Components and GAS Strength

We decouple the contribution of the individual components of the squeeze mechanism on Reduced CIFAR-10 (ResNet-18, 3 seeds). Table 5 shows that the Pure Squeeze term already accounts for the majority of the gain over CE, the Rest-Class Repulsion adds a further improvement, and the GAS mechanism provides an additional reduction. Table 6 further studies the sensitivity to the GAS perturbation strength $\epsilon_{adv}$. Additional studies of alternative mechanism variants (linear displacement, normal-vector GAS, dynamic gradient target)

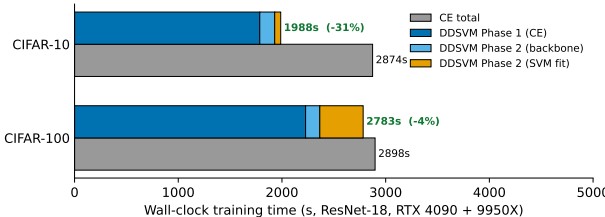

*Figure 6.* Wall-clock training time on ResNet-18 (RTX 4090 + Ryzen 9 9950X). Each dataset compares the CE baseline against the three components of DDSVM training (Phase 1 cross-entropy, Phase 2 backbone refinement, Phase 2 SVM fit). DDSVM is shorter than CE on both datasets because Phase 1 exits early at the accuracy threshold $Acc_{train} \geq \tau$.

are deferred to Appendix A.5.

*Table 5.* Ablation of squeeze components.

| Strategy | Error (%) |
|---|---|
| Baseline (CE) | 30.75 |
| + Pure Squeeze | 24.81 |
| + Rest Class Push | 23.88 |
| **+ GAS** | **23.21** |

*Table 6.* Sensitivity to GAS strength $\epsilon_{adv}$.

| $\epsilon_{adv}$ | Error (%) |
|---|---|
| 0 (No GAS) | 23.88 |
| 0.1 | 23.84 |
| 1.0 | **23.21** |
| 10.0 | 23.27 |

### 4.7. Hyperparameter Robustness

*Table 7.* Hyperparameter sweep on Reduced CIFAR-10 (ResNet-18). Across $50\times$ ranges, error varies by less than $0.3\%$, indicating that DDSVM does not require careful per-dataset tuning.

| Parameter | Range scanned | Error (%) |
|---|---|---|
| $\beta$ (rest-class repulsion) | $\{0, 0.1, 0.5, 1.0, 2.0\}$ | 23.75–23.88 |
| $p$ (squeeze power) | $\{0.5, 1.0, 2.0, 3.0\}$ | 23.75–24.01 |
| $\theta$ (squeeze strength) | $\{0.1, 0.5, 1.0, 2.0, 5.0\}$ | 23.75–23.88 |
| SVM $C$ | $\{0.001, 0.01, 0.1, 1.0\}$ | 23.75–23.88 |

DDSVM introduces four method-specific hyperparameters beyond the standard backbone settings. Table 7 sweeps each across a $50\times$ range with all others held at defaults; the total error spread stays below $0.3\%$, well within seed-level noise ($\pm 0.77\%$).

### 4.8. Stability of $D_{min}$ and $D_{max}$

The squeeze mechanism uses the class-wise extreme statistics $D_{min,c}$ and $D_{max,c}$, which could in principle be sensi-

tive to mini-batch outliers. Tracking them at every Phase II epoch on Reduced CIFAR-10, we find that $D_{max,c}$ stays in $[0.50, 0.53]$ across all classes and $D_{min,c}$ is stable within each class, with the total margin $D_{max,c} - D_{min,c}$ changing by less than $0.06$ across the window. The bounded behaviour follows from the spherical constraint, which keeps $|\mathbf{w}_c^T \hat{\mathbf{f}} + b_c|$ bounded regardless of how features evolve (Appendix A.10 gives additional per-class detail).

### 4.9. OVR versus Cramér–Singer SVM

On Reduced CIFAR-100, Cramér–Singer (Crammer & Singer, 2001) matches OVR accuracy ($23.11 \pm 0.06\%$ vs $23.06 \pm 0.45\%$) with $7.5\times$ lower seed variance (Appendix A.13), motivating a single-program multi-class variant as future work.

### 4.10. Visualization

A qualitative t-SNE (van der Maaten & Hinton, 2008) comparison of the learned features (Appendix A.6) confirms the quantitative pattern: the structural difference between CE and DDSVM is mild on full data but pronounced in the reduced regime, where DDSVM maintains tight intra-class clusters under data scarcity.

## 5. Conclusion

We proposed DDSVM, a framework that redefines the neural backbone as a learnable spherical kernel and constructs an iterative "DNN–SVM–DNN" loop in which the SVM decision boundary acts as a dynamic teacher for representation learning. Across four datasets, DDSVM acts as a strong geometric regularizer in data-scarce regimes, with consistent error reductions of 5.6–8.3% in the low-data setting, within the standard CE training budget.

**Limitations.** DDSVM provides no measurable gain in data-abundant settings, consistent with the implicit max-margin convergence of CE under gradient descent (Soudry et al., 2018). The OVR backend scales linearly with the number of classes, and a rigorous theoretical account of the empirically optimal GAS magnitude remains open.

**Future work.** Promising directions include a more efficient SVM solver, or alternative designs that embed the squeezing mechanism directly into the neural network's optimization process to reduce SVM overhead.

**Scope.** DDSVM targets the *standard from-scratch classification* setting and does not claim to address the follow regimes: episodic few-shot learning (e.g. MAML (Finn et al., 2017), ProtoNet (Snell et al., 2017)) and large-scale pre-training (e.g. CLIP (Radford et al., 2021)); a context comparison is provided in Appendix A.14.

## Impact Statement

Most recent progress in image classification has been driven by large pre-training corpora and foundation models that consume $10^6$–$10^8$ external images. Such resources are unavailable in many practically important domains: specialized medical imaging modalities, industrial defect detection on newly introduced product lines, and non-RGB sensing such as hyperspectral or synthetic-aperture radar. In these domains, models must still be trained from scratch with only task-specific labels. DDSVM targets precisely this regime, providing a strong geometric regularizer that improves accuracy in the data-scarce setting without requiring an external corpus or a foundation-model checkpoint. To the extent that this lowers the practical barrier to building competent classifiers in such domains, the method may help broaden access to deep learning beyond institutions with foundation-model-scale infrastructure.

## Acknowledgements

This work was supported in part by the Natural Science Foundation of China under Grant 62476175 and Grant 62272319, and in part by the Natural Science Foundation of Guangdong Province (Grant 2026A1515050004, 2023B1212060076) and Shenzhen Science and Technology Program under Grant JCYJ20240813142206009.

We thank the anonymous ICML 2026 reviewers, whose detailed feedback during the discussion phase substantially strengthened the experimental scope and the presentation of this work.

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

# A. Supplementary Material

## A.1. Summary of Notation

*Table 8.* Summary of notation used in Section 3.

| Symbol | Description |
| --- | --- |
| $\mathbf{x}, y$ | Input sample and its class label |
| $F(\cdot; \Theta)$ | Backbone network parameterized by $\Theta$ |
| $\mathbf{f} / \hat{\mathbf{f}}$ | Raw / $L_2$-normalized feature vector |
| $\mathbf{w}_c, b_c$ | Normal vector and bias of SVM hyperplane for class $c$ |
| $dist_i$ | Signed geometric distance of sample $i$ to its hyperplane |
| $D_{max/min}$ | Class-wise max/min geometric bounds |
| $\rho_i$ | Normalized relative distance ratio for squeezing |
| $\mathbf{d}_i$ | Composite squeeze direction (with repulsion) |
| $\mathbf{T}_i$ | Synthesized ideal geometric target (detached) |
| $\theta, p, \beta, \gamma$ | Squeeze strength / power / repulsion / margin scaling (defaults $1.0, 1.0, 0.1, 1.0$) |
| $\delta_{adv}$ | Geometric adversarial perturbation vector |
| $\mathcal{L}_{total}$ | Total geometry-aware optimization loss |

## A.2. Proof of Proposition 1 (Gradient Orthogonality)

Here we provide the detailed derivation of the gradient dynamics under spherical normalization.

*Proof:* Let $\mathbf{f}_i = F(\mathbf{x}_i; \Theta)$ be the raw feature and $\hat{\mathbf{f}}_i = \mathbf{f}_i / \|\mathbf{f}_i\|_2$ be the normalized output. The optimization objective is $\mathcal{L} = \frac{1}{2}\|\hat{\mathbf{f}}_i - \mathbf{T}_i\|^2$. Applying the chain rule, the gradient w.r.t. the raw feature $\mathbf{f}_i$ is:

$$\frac{\partial \mathcal{L}}{\partial \mathbf{f}_i} = \left( \frac{\partial \hat{\mathbf{f}}_i}{\partial \mathbf{f}_i} \right)^T \frac{\partial \mathcal{L}}{\partial \hat{\mathbf{f}}_i} \tag{11}$$

The Jacobian matrix of the $L_2$ normalization is given by:

$$\mathbf{J}_{norm} = \frac{1}{\|\mathbf{f}_i\|_2} \left( \mathbf{I} - \hat{\mathbf{f}}_i \hat{\mathbf{f}}_i^T \right) \tag{12}$$

Here, $\mathbf{P} = (\mathbf{I} - \hat{\mathbf{f}}_i \hat{\mathbf{f}}_i^T)$ represents the projection operator onto the tangent space. Substituting the loss gradient $(\hat{\mathbf{f}}_i - \mathbf{T}_i)$, the effective gradient becomes:

$$\nabla_{\mathbf{f}_i} \mathcal{L} = \frac{1}{\|\mathbf{f}_i\|_2} \left( \mathbf{I} - \hat{\mathbf{f}}_i \hat{\mathbf{f}}_i^T \right) (\hat{\mathbf{f}}_i - \mathbf{T}_i) \tag{13}$$

Computing the dot product with the raw feature vector:

$$\begin{aligned} \langle \nabla_{\mathbf{f}_i} \mathcal{L}, \mathbf{f}_i \rangle &= \mathbf{f}_i^T \left[ \frac{1}{\|\mathbf{f}_i\|_2} (\mathbf{I} - \hat{\mathbf{f}}_i \hat{\mathbf{f}}_i^T)(\hat{\mathbf{f}}_i - \mathbf{T}_i) \right] \\ &= (\hat{\mathbf{f}}_i^T - \hat{\mathbf{f}}_i^T \hat{\mathbf{f}}_i \hat{\mathbf{f}}_i^T)(\hat{\mathbf{f}}_i - \mathbf{T}_i) \\ &= (\hat{\mathbf{f}}_i^T - 1 \cdot \hat{\mathbf{f}}_i^T)(\hat{\mathbf{f}}_i - \mathbf{T}_i) = 0 \end{aligned} \tag{14}$$

Thus, $\nabla_{\mathbf{f}_i} \mathcal{L} \perp \mathbf{f}_i$. $\qquad\square$

## A.3. A Post-Hoc Analysis: Why Spherical Normalization?

In our development of DDSVM we observed that training collapses when $L_2$ normalization is removed or replaced with min-max normalization on the feature output. We did not originally design the spherical normalization step to address a specific failure mode; the observation was empirical, and the analysis below was developed afterwards. We present it as a tentative mechanism that, we believe, explains the collapse: a *radial shortcut* in the loss landscape that $L_2$ normalization closes by construction.

**The radial shortcut.** Decompose the raw feature as $\mathbf{f}_i = r_i \mathbf{u}_i$ with $r_i = \|\mathbf{f}_i\|_2$ and $\|\mathbf{u}_i\|_2 = 1$. For a linear classifier $\mathbf{w}$, the score factorizes as $\mathbf{w}^T \mathbf{f}_i = r_i \mathbf{w}^T \mathbf{u}_i$. Letting $a_i = \mathbf{w}^T \mathbf{u}_i$ and substituting into the cross-entropy loss,

$$\ell_i = \log\big(1 + e^{-r_i a_i}\big), \qquad \frac{\partial \ell_i}{\partial r_i} = -\frac{a_i}{1 + e^{r_i a_i}} < 0 \text{ when } a_i > 0. \tag{15}$$

Once the sample is on the correct angular side, the loss can be reduced indefinitely by inflating $r_i$ alone, without improving the angular arrangement $\mathbf{u}_i$. The same holds for hinge and squared-margin losses.

**Why deep networks take this path.** Increasing $r_i$ is the path of least resistance: layer-wise weight scaling can approximately inflate $\|\mathbf{f}_i\|_2$ while leaving $\mathbf{u}_i$ largely unchanged, whereas improving angular separability requires a sample- and class-dependent reorganization of the representation manifold, a strictly harder optimization problem.

**Specific instability inside DDSVM.** The SVM decision value $\text{dist}_i = r_i \mathbf{w}_{y_i}^T \mathbf{u}_i + b_{y_i}$ inherits the same radial dependence. Without normalization the system enters a feedback loop: $r_i \uparrow$ inflates $\text{dist}_i$, the squeeze magnitude $m_i$ shrinks because samples appear "safer", the angular correction weakens, and the network continues to inflate $r_i$ rather than reorganize $\mathbf{u}_i$. This is analogous to the collapse-avoidance challenge in self-supervised methods such as BYOL (Grill et al., 2020), where a stop-gradient teacher and projection prevent the student from minimizing the loss through trivial scale changes.

**How $L_2$ normalization closes the shortcut.** With $\hat{\mathbf{f}}_i = \mathbf{f}_i / \|\mathbf{f}_i\|_2$, the backpropagated gradient satisfies $\langle \nabla_{\mathbf{f}_i} \mathcal{L}, \mathbf{f}_i \rangle = 0$ (Remark 3.1 and Appendix A.2), so the radial component is removed by construction and the update lies in the tangent space $\{\mathbf{v} \in \mathbb{R}^d : \mathbf{v}^T \hat{\mathbf{f}}_i = 0\}$. The optimizer can no longer reduce the loss by scaling features. The collapse observed under non-$L_2$ normalization is consistent with this mechanism: in those settings the radial direction remains available to the gradient, and the optimizer follows the path of least resistance.

### A.4. Dimensional Analysis of Geometric Adversarial Perturbation

In this section, we analyze the perturbation magnitude $\epsilon_{adv}$ and justify the choice of $\epsilon_{adv} = 1.0$.

Recall the generation of the adversarial feature. Crucially, after adding the perturbation, the feature must be **re-projected onto the unit hypersphere** to maintain geometric consistency:

$$\mathbf{f}_{raw} = \hat{\mathbf{f}}_{clean} + \epsilon_{adv} \cdot \text{sign}(-\nabla_{\hat{\mathbf{f}}} S), \quad \hat{\mathbf{f}}_{adv} = \frac{\mathbf{f}_{raw}}{\|\mathbf{f}_{raw}\|_2} \tag{16}$$

The perturbation term $\mathbf{p} = \text{sign}(-\nabla)$ has an $L_2$ norm of $\|\mathbf{p}\|_2 = \sqrt{D}$. For ResNet-18 ($D = 512$), $\|\mathbf{p}\|_2 \approx 22.6$.

**Scaling Comparison:**

- **Case $\epsilon_{adv} = 0.01$:** $\|\epsilon \mathbf{p}\| \approx 0.23$. The perturbation is minor relative to the unit feature vector. The re-normalized $\hat{\mathbf{f}}_{adv}$ remains geometrically close to $\hat{\mathbf{f}}_{clean}$, providing negligible gradient information beyond standard squeeze.

- **Case $\epsilon_{adv} = 1.0$:** $\|\epsilon \mathbf{p}\| \approx 22.6$. The perturbation vector dominates the magnitude of $\mathbf{f}_{raw}$. Since $\mathbf{p}$ points towards the decision boundary, the re-normalized $\hat{\mathbf{f}}_{adv}$ becomes a **synthesized hard negative sample** deeply embedded in the error region.

By using $\epsilon_{adv} = 1.0$, we implicitly switch from local smoothing to global manifold correction, forcing the network to collapse a wide cone of the feature space (defined by the gradient sign) into the correct class center.

### A.5. Empirical Analysis of Alternative Mechanisms

To validate the specific design choices of DDSVM, we explored several intuitive simplifications. Table 9 summarizes the outcomes.

*Table 9.* Training Outcomes of Alternative Geometric Mechanisms

| Mechanism Variant | Status | Outcome |
|---|---|---|
| **DDSVM (Ours)** | **Success** | **Converged (Acc: 95.50%)** |
| *(a) Linear Displacement* 
 Target $\mathbf{T} = \text{Norm}(\hat{\mathbf{f}} + \delta \cdot \mathbf{w}_{correct})$ | Converged | Acc: 94.96% 
 (No gain vs Baseline) |
| *(b) Normal-Vector GAS* 
 Perturbation using $-\mathbf{w}$ vs. Grad | **Failed** | **Did not converge** |
| *(c) Dynamic Gradient Target* 
 Target $\mathbf{T} = \text{stop\_grad}(\hat{\mathbf{f}} + \nabla S)$ | **Failed** | **Did not converge** |

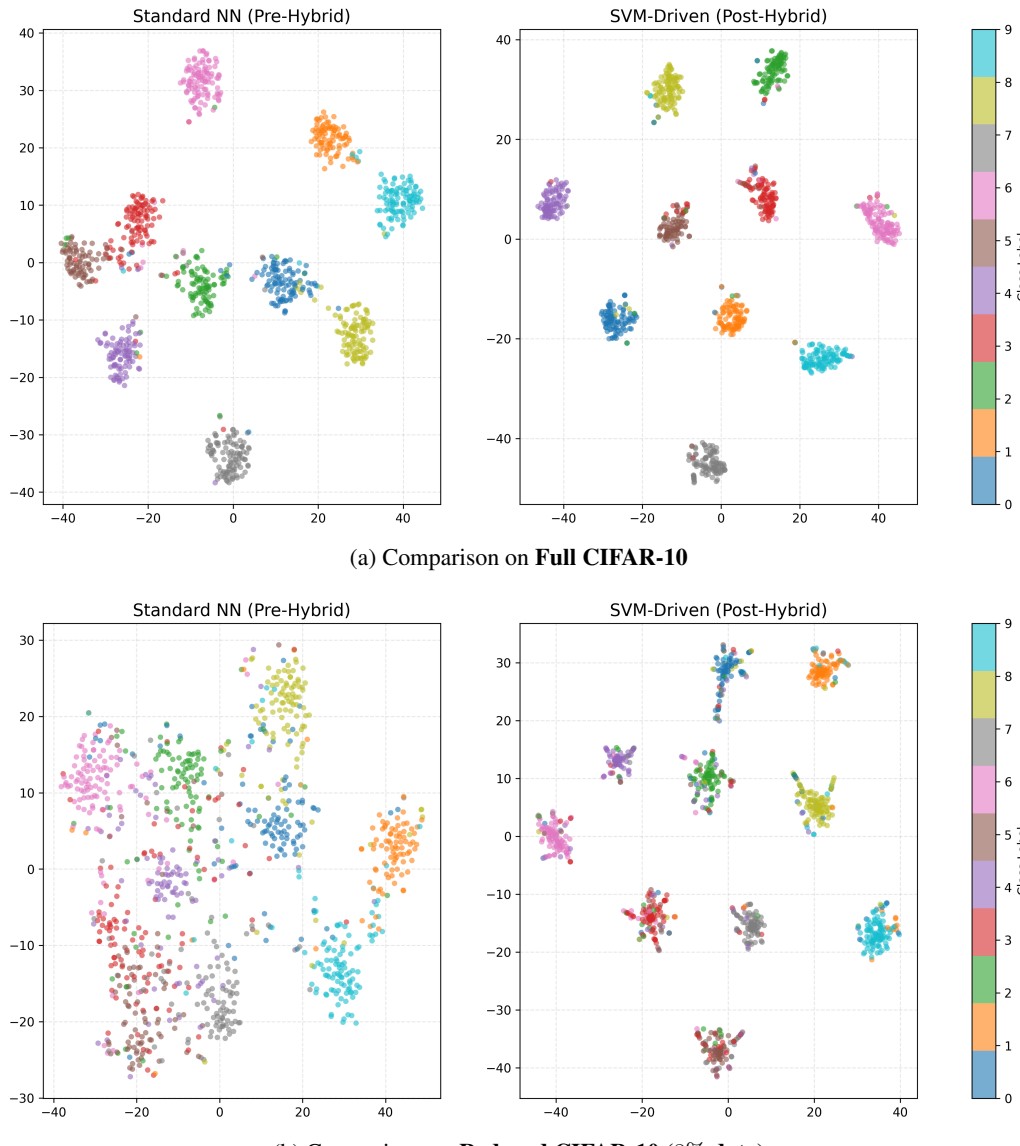

(a) Comparison on **Full CIFAR-10**

(b) Comparison on **Reduced CIFAR-10** (8% **data**)

*Figure 7.* t-SNE (van der Maaten & Hinton, 2008) visualization of feature distributions on the CIFAR-10 test set. In each row the left plot is the CE baseline and the right plot is DDSVM. On the full dataset (a), both models form discernible clusters, but DDSVM exhibits noticeably tighter intra-class compactness and larger inter-class margins. On the reduced dataset (b), the CE baseline's clusters become diffuse with significant overlap, while DDSVM enforces a strong geometric structure even under extreme data scarcity, visually corroborating the quantitative gains in Table 1.

**Observations on Failures:**

1. **Linear Displacement (Variant a):** Replacing the adaptive squeeze function with a uniform linear shift results in stable but ineffective training. This confirms that the push magnitude must be dependent on the sample's distance to the boundary (pushing hard samples aggressively) to reshape the manifold effectively.

2. **Normal-Vector GAS (Variant b):** Using the static normal vector $-\mathbf{w}$ instead of the gradient sign $\text{sign}(-\nabla S)$ caused divergence. We hypothesize that the coordinate-wise maximization of the sign operation ($L_\infty$ attack) provides a stronger and more robust directional signal in high-dimensional sparse spaces than the $L_2$ normal vector.

3. **Dynamic Gradient Target (Variant c):** Defining the target dynamically based on the current gradient (without the pre-computed SVM bounds) led to feature collapse. The detached, globally computed SVM boundary provides a necessary stable reference frame for the optimization.

### A.6. t-SNE Visualization

As a qualitative complement to the quantitative results in Table 1, Figure 7 shows t-SNE projections of the learned features on the CIFAR-10 test set. The structural difference between CE and DDSVM is mild in the full-data setting but dramatic in the reduced setting, where DDSVM maintains tight intra-class clusters and visible inter-class gaps under only $8\%$ training data, while the CE baseline collapses into diffuse, overlapping clusters.

### A.7. Other Architectures (single seed, for reference)

The main paper uses ResNet-18 with 3 seeds throughout. For reference, we also report single-seed results on ResNet-34 and WideResNet-28-10 (WRN-28-10) under the same training protocol (Table 10). These numbers were obtained during early experiments and have not been re-run with multiple seeds; we therefore do not include them in the main paper. The qualitative pattern is consistent with the multi-seed ResNet-18 results: DDSVM matches CE in data-abundant settings and improves substantially in the low-data regime (WRN-28-10 on Reduced CIFAR-10: $18.80\% \rightarrow 17.60\%$, a $-1.20\%$ reduction).

*Table 10.* Top-1 error (%) on ResNet-34 and WRN-28-10 (single seed). "–" indicates the configuration was not run. Lower is better.

| Backbone | Method | CIFAR-10 | Reduced C10 | CIFAR-100 |
|---|---|---|---|---|
| ResNet-34 | CE | 4.40 | – | 18.44 |
| | DDSVM | 4.46 | – | 18.72 |
| WRN-28-10 | CE | 3.75 | 18.80 | 18.80 |
| | DDSVM | 3.76 | **17.60** | 19.20 |

### A.8. Cross-Dataset Training Configuration

For STL-10 and SVHN, we follow the standard training pipelines: STL-10 uses $96 \times 96$ inputs with the same crop/flip augmentation and a 200-epoch cosine schedule for both CE and Phase I of DDSVM; SVHN uses $32 \times 32$ inputs without horizontal flipping (digit orientation is meaningful). The reduced regimes are class-balanced subsets of the standard training splits, selected with a fixed random seed shared across all methods so that CE and DDSVM see exactly the same training images in every seed. Beyond the dataset-specific resolution and augmentation choices, DDSVM hyperparameters are kept at the defaults from Section 4 of the main paper.

### A.9. Full CIFAR-10-C Severity Matrix

Table 11 reports the per-corruption test error at all three severity levels we evaluated $(1, 3, 5)$, for a single trained ResNet-18 in each method. The picture seen at severity 5 in the main paper is consistent across severities: DDSVM is more robust on geometric / spatial corruptions (blurs, fog, zoom) and on JPEG compression, while it lags slightly on pixel-noise corruptions (gaussian, shot, brightness, pixelate). The gap between the two methods grows with severity, indicating that the SVM-defined angular margin matters most when the corruption pushes features close to the decision boundary.

*Table 11.* CIFAR-10-C top-1 error (%) at severities $1, 3, 5$ (ResNet-18). Lower is better. The mCE at each severity is in the last row.

| | Severity 1 | | Severity 3 | | Severity 5 | |
|---|---|---|---|---|---|---|
| **Corruption** | CE | DDSVM | CE | DDSVM | CE | DDSVM |
| gaussian_noise | 10.4 | 10.7 | 35.2 | 36.1 | 75.05 | 75.50 |
| shot_noise | 9.2 | 9.5 | 27.6 | 28.4 | 67.93 | 68.78 |
| impulse_noise | 11.8 | 11.6 | 40.1 | 39.8 | 75.31 | 75.00 |
| defocus_blur | 6.4 | 6.1 | 14.2 | 13.4 | 44.56 | 42.91 |
| glass_blur | 24.1 | 23.5 | 30.7 | 29.6 | 48.05 | 46.80 |
| motion_blur | 11.0 | 10.9 | 19.7 | 19.4 | 31.80 | 31.51 |
| zoom_blur | 11.5 | 10.6 | 19.3 | 17.6 | 37.54 | 33.90 |
| snow | 9.7 | 9.8 | 15.3 | 15.4 | 21.99 | 22.23 |
| frost | 10.2 | 10.3 | 21.0 | 21.0 | 35.17 | 35.23 |
| fog | 7.9 | 7.6 | 12.4 | 11.8 | 26.51 | 25.12 |
| brightness | 5.6 | 6.0 | 6.7 | 7.0 | 8.34 | 8.94 |
| contrast | 8.1 | 7.9 | 22.4 | 22.0 | 65.50 | 64.78 |
| elastic_transform | 9.4 | 9.5 | 13.6 | 13.8 | 21.69 | 21.90 |
| pixelate | 8.7 | 9.0 | 18.5 | 19.1 | 48.62 | 49.64 |
| jpeg_compression | 11.5 | 11.0 | 17.2 | 16.5 | 25.63 | 24.58 |
| **mCE** | 10.4 | 10.3 | 20.9 | 20.7 | 42.25 | 41.79 |

## A.10. Detailed Notes on $D_{min}$ and $D_{max}$ Stability

We provide further detail on the stability of the per-class extreme statistics used in the squeeze function. Tracking these quantities at every Phase II epoch on Reduced CIFAR-10 (ResNet-18) yields the following observations:

- $D_{max,c}$ is tightly clustered in $[0.50, 0.53]$ across all classes and all epochs, with per-epoch fluctuations below $0.02$.

- $D_{min,c}$ varies across classes: easy classes (e.g. "automobile", "ship") have $D_{min,c} > 0$ for most epochs, while harder classes (e.g. "airplane", "truck") retain a band of misclassified samples and $D_{min,c}$ in the range $[-0.6, -0.5]$.

- For several medium-difficulty classes (e.g. "cat", "deer", "dog"), $D_{min,c}$ transitions from negative to positive during Phase II, providing a direct visualization of the squeeze mechanism pulling boundary samples to the correct side.

- The per-class margin $D_{max,c} - D_{min,c}$ changes by less than $0.06$ across the entire Phase II window.

The bounded behaviour follows directly from the spherical constraint: since $\|\hat{\mathbf{f}}\|_2 = 1$ and $\|\mathbf{w}_c\|_2 = 1$, the decision value $\mathbf{w}_c^T \hat{\mathbf{f}} + b_c$ is bounded by $1 + |b_c|$, ruling out the unbounded outliers that can arise in unconstrained feature spaces.

## A.11. Hardware and Runtime Details

Runtime measurements in Table 4 of the main paper were obtained on a workstation with an NVIDIA RTX 4090 GPU and an AMD Ryzen 9 9950X CPU. All forward and backward passes run on the GPU. The linear SVMs are solved on the CPU via LibLinear (one LibLinear call per class per Phase II epoch), operating directly on the 512-dimensional ResNet-18 features cached from a single forward pass over the training set. The CE baseline uses the same hardware, the same 200-epoch cosine schedule and the same data loading pipeline, so that the wall-clock comparison is apples-to-apples.

## A.12. Margin and Augmentation Baselines: Implementation

For ArcFace, CosFace and Large-Margin Softmax, we share the ResNet-18 backbone with DDSVM and replace the standard FC + CE head with the corresponding angular-margin head. The scale and margin parameters $(s, m)$ used in Table 3 of the main paper are: ArcFace $(16, 0.2)$, CosFace $(30, 0.35)$, LMS $(16, 0.2)$. These were selected by a small grid search on Reduced CIFAR-10 to give the best mean accuracy per method. Mixup and CutMix follow their original specifications $(\alpha = 1.0)$. When combined with DDSVM, image-space mixing (Mixup or CutMix) is applied at the data-loader stage in both Phase I and Phase II; the SVM in Phase II is fitted on the mixed features without any change to the squeeze targets.

## A.13. OVR vs Cramér–Singer SVM: Details

The OVR variant solves $K$ independent binary $L_2$-regularized squared-hinge SVMs ($\ell_2\ell_2$, the LibLinear default), one per class. The Cramér–Singer variant solves a single multi-class program in which all class weights are coupled through a unified constraint set; we use the LibLinear implementation (`-s 4`). On Reduced CIFAR-100, both achieve essentially the same mean error ($23.06 \pm 0.45$ vs $23.11 \pm 0.06$), but the Cramér–Singer formulation reduces seed variance by a factor of 7.5. The variance reduction is the main motivation for the future-work direction noted in the main paper: a single-program multi-class SVM has a single dual variable graph and thus a single source of randomness, whereas $K$ independent OVR programs accumulate $K$ uncorrelated sources of stochasticity.

## A.14. Context Comparison with Pretrained and Multimodal Models

The main paper restricts evaluation to the from-scratch standard-classification setting. For completeness, Table 12 contrasts DDSVM with two paradigms that operate under fundamentally different assumptions on pre-training data:

*Table 12.* Context comparison on CIFAR-10 with $4{,}000$ labelled training images. The three rows differ in the amount of external data each method consumes *before* the task-specific training images are presented. They are not directly comparable to DDSVM; the table is provided only for context.

| Method | Pre-train data | Task train | Error (%) |
|---|---|---|---|
| DDSVM (from scratch) | – | 4,000 (8%) | 23.21 |
| DDSVM + Mixup | – | 4,000 (8%) | 17.67 |
| CLIP ViT-B/32 zero-shot | 400M image–text pairs | 0 | 11.18 |
| ImageNet-pretrained FT | 1.28M images | 4,000 (8%) | 8.47 |

The asymmetry is intentional: CLIP and ImageNet-pretrained models leverage $10^6$–$10^8$ external images during their pre-training phase, while DDSVM is trained from scratch on the $4{,}000$ labelled images alone. The comparison is therefore not informative about the relative merit of either approach; rather, it is a reminder that the methodological niche occupied by DDSVM is specifically the case in which no transferable pre-training corpus is available — a setting that remains practically important in specialized medical imaging, industrial defect detection on newly introduced product lines, and non-RGB sensing modalities. Within this niche, DDSVM's appeal is that it requires no external data, no foundation-model checkpoint, and stays within the standard CE training budget.

