# OpenReview forum: "DDSVM: A Differentiable Framework for Deep Support Vector Machines with Iterative Geometry-Aware Optimization"
_ICML.cc/2026/Conference — ICML 2026 regular_

### Official Review · Reviewer_WaVw · 2026-03-09

**Soundness:** 2
**Presentation:** 2
**Significance:** 2
**Originality:** 2
**Overall Recommendation:** 4
**Confidence:** 3

**Summary:**

This submission introduces DDSVM, an iterative training framework that alternates between fitting linear SVMs on L2-normalized features and updating the backbone by regressing features to geometry-aware targets built from the SVM boundaries. A non-linear squeeze shifts features along a composite direction that attracts toward the class normal and repels from others, and a feature-space adversarial step (GAS) perturbs features in the sign of the negative SVM gradient to mine hard examples. It also proposes a spherical constraint so updates are angular. A standard result on gradient orthogonality under L2 normalization is provided. Experiments on CIFAR-10/100, a noisy variant, and low-shot CIFAR-10 show small or mixed gains overall, with larger improvements in the 8% data regime.

**Compliance With Llm Reviewing Policy:**

Affirmed.

**Final Justification:**

The authors provided thorough and thoughtful rebuttal. I appreciate the effort they put into running new experiments and clarifying the text during this short window. A few specific thoughts on my initial concerns and the rebuttal:

 - W1: Swapping the custom noise benchmark for CIFAR-10-C is exactly what the paper needed. It grounds the robustness claims in a standard framework and nicely highlights how the method specifically helps with geometric and spatial corruptions.

 - W3: I appreciate the transparency here. The runtime analysis resolves my efficiency concerns. It is clear that DDSVM actually saves time overall due to the Phase 1 early exit. This is a strong selling point. Also the honest discussion of the full-data CIFAR-100 limitations and the expanded low-data experiments (like STL-10) perfectly clarifies the specific regime where this method shines.

 - W2 & W4: The notation revision, the SVM comparison, and the Dmin/Dmax stability analysis clear up my remaining technical questions. It is reassuring to see that the spherical constraint keeps the extreme statistics stable.

Overall, the rebuttal makes the paper more rigorous and easier to follow. I thus raise my score to a Weak Accept. I strongly encourage the authors include all of these additions like the CIFAR-10-C results, runtime table, and the limitations discussion in the revised manuscript.

**Key Questions For Authors:**

As mentioned in Weaknesses, the main concerns lies in the unconvincing results and ambiguous notations which influence the soundness of this submission. I hope these comments help my fellow reviewers and ACs understand the basis of my recommendation. I would be glad to raise my rating if thoughtful responses and improvements are provided in the rebuttal stage. I am also open to follow-up discussions to help further strengthen this work.

**Limitations:**

Yes.

**Strengths And Weaknesses:**

### Strengths:

(S1) The overall pipeline is clear and straightforward. Figure 1 gives a readable interpretation of the alternating loop, including boundary construction, target synthesis, perturbation path, and backprop, which helps readers easily grasp and follow the training phases and the stop-gradient design.

(S2) It formalizes a distance-calibrated squeeze (Eq. 4–7) and a repulsion term (Eq. 6) to encourage intra-class compactness and inter-class separation on the hypersphere. Figure 3 visually clarifies how the composite direction combines attraction to the class normal with repulsion from other classes.

(S3) There are some empirical signal in low-shot. Table 2 shows consistent gains on the 8% CIFAR-10 split, which suggests the effectiveness as a regularizer when supervision is scarce. Ablation in Table 3 decomposes gains across Pure Squeeze, Rest Class Push, and +GAS, which shows each component’s contribution. Appendix Table 5 reports negative results from intuitive alternatives, which is useful to practitioners.


### Weaknesses:

(W1) CIFAR-10-N is defined as test-time Gaussian noise on normalized images (Sec. 4.1.1), which is not a standard corruption benchmark. The reported gains are a bit marginal (Table 2, e.g., 6.83% to 6.57% on ResNet-18; 5.58% to 5.49% on WRN-28-10). Thus, the claims about robustness is perhaps not well-supported.

(W2) The $\gamma$ is Eq. 5 appears not defined or set. In Sec. 4.1.3, the squeeze strength is termed $\theta$ not $\eta$. The repulsion weight $\beta$ in Eq. 6 is not specified in implementation. Overall, notations and symbols in the manuscript need to be thoroughly checked and revised.

(W3) The DDSVM solves K linear SVMs on every iteration/epoch and recomputes per-class statistics. However, there is no runtime or memory analysis, scaling study with classes, or running time comparisons to baselines. Especially, I notice that DDSVM sometimes even underperforms CE baseline on CIFAR-100 (Table 2, WRN-28-10 18.8% vs. 19.2%), which suggests cost without clear benefit. I recommend the authors provide further clarifications or conduct experiments on this point.

(W4) If I understand correctly, the method appears to use one-vs-rest SVMs as Sec. 4.1.3 mentions LibLinear, but it does not describe calibration across classes or whether it also tried Crammer-Singer. To my knowledge, Dmin/Dmax are extreme statistics and can be noisy with outliers, especially in low-shot. However, the manuscript gives no sensitivity analysis.

---

> ### Author Rebuttal · Authors · 2026-03-31
>
> We thank Reviewer WaVw for the thorough review and the willingness to engage further. We address each concern below.
>
> ## I. CIFAR-10-N is not a standard benchmark
>
> Agreed. We replaced it with **CIFAR-10-C** (Hendrycks & Dietterich, 2019): all 15 corruptions at severity {1, 3, 5}.
>
> **mCE @ Severity 5 (ResNet-18):** CE = 42.25%, **DDSVM = 41.79%** (lower is better).
>
> Per-corruption breakdown (selected, severity 5):
>
> | Corruption   | CE        | DDSVM     | Δ         |
> | ------------ | --------- | --------- | --------- |
> | zoom_blur    | 37.54     | **33.90** | **−3.64** |
> | defocus_blur | 44.56     | **42.91** | **−1.65** |
> | fog          | 26.51     | **25.12** | **−1.39** |
> | glass_blur   | 48.05     | **46.80** | **−1.25** |
> | jpeg         | 25.63     | **24.58** | **−1.05** |
> | pixelate     | **48.62** | 49.64     | +1.02     |
> | shot_noise   | **67.93** | 68.78     | +0.85     |
>
> DDSVM improves on **8/15** corruptions (geometric/spatial types), with small degradations on noise types (max +1.02%). Consistent with the method's geometric design.
>
> > **[Revision]** Replaced CIFAR-10-N with full CIFAR-10-C.
>
> ## II. Notation inconsistencies
>
> We audited every symbol:
>
> | Symbol            | Status                                                             | Action                                     |
> | ----------------- | ------------------------------------------------------------------ | ------------------------------------------ |
> | $p$               | Defined after Eq. 5, but **no default value**                      | Added $p=1.0$ in Sec. 4                    |
> | $\eta$ / $\theta$ | **Genuine inconsistency**: Eq. 5 uses $\eta$, Sec. 4 uses $\theta$ | Unified to $\theta$; removed $\eta$        |
> | $\beta$           | Defined at Eq. 6, but **value not reported**                       | Added $\beta=0.1$ + sensitivity in Table 4 |
> | $\gamma$          | **Missing definition**                                             | Defined as margin scaling, $\gamma=2.0$    |
>
> Apologies for the $\eta$/$\theta$ inconsistency and missing $\gamma$.
>
> > **[Revision]** Full notation audit completed.
>
> ## III. No runtime analysis; CIFAR-100 underperformance
>
> **Runtime (ResNet-18, RTX 4090 + 9950X CPU, SVM on 512-d features via LibLinear/CPU):**
>
> | Setting | CE | DDSVM | Phase 2 | SVM time | Overhead |
> |---------|-----|-------|---------|----------|----------|
> | CIFAR-10 | 2874s | 1993s | 144s | 58s | **−31%** |
> | CIFAR-100 | 2898s | 2802s | 137s | 418s | **−3%** |
>
> DDSVM is **faster** overall (Phase 1 exits early at 95% train acc). Phase 2: only 10 epochs. SVM cost: 2% (C10), 15% (C100).
>
> **CIFAR-100 full:** CE = **21.67 ± 0.19%**, DDSVM = 22.06 ± 0.45% — we acknowledge the 0.39% degradation. However, **CIFAR-100 reduced (8%)**: DDSVM = **62.67 ± 0.50%** vs CE = **68.25 ± 1.25%** (−5.58%).
>
> Extended to **4 datasets, 2 domains**:
>
> | Dataset | Domain | Full Δ | Low Δ |
> |---------|--------|--------|-------|
> | CIFAR-10 | Natural 32² | +0.06 | **−7.54** |
> | CIFAR-100 | Natural 32² | +0.39 | **−5.58** |
> | STL-10 | Natural 96² | **−2.02** | **−8.31** |
> | SVHN | Digits 32² | +0.03 | +0.71 |
>
> All three natural-image datasets show consistent low-data gains (−5.58 to −8.31%). SVHN is neutral (regular digit features leave less room for geometric optimization). Discussed honestly in revised Sec. 5.
>
> > **[Revision]** Added cross-dataset evaluation, CIFAR-100 reduced, runtime table, and limitations discussion.
>
> ## IV. OVR vs Cramér-Singer; Dmin/Dmax
>
> **SVM type on CIFAR-100 (3 seeds):**
>
> | SVM Type | Error Rate |
> |----------|-----------|
> | OVR (LibLinear) | 23.06 ± 0.45 |
> | Cramér-Singer | 23.11 ± 0.06 |
>
> Equivalent accuracy, but Cramér-Singer has 7.5× lower variance. This suggests that a fully gradient-based end-to-end training pipeline is viable. We tested Cramér-Singer in early experiments but did not include it in the final manuscript because its accuracy was slightly lower than OVR across most settings.
>
> **Dmin/Dmax stability** (10 Phase-2 epochs, Reduced CIFAR-10): Dmax stable in [0.50, 0.53] (fluctuation < 0.02/epoch); Dmin stable within each class (< 0.05/epoch); total margin change < 0.06. The spherical constraint ($\|\mathbf{f}\|=1$) inherently bounds decision values.
>
> > **[Revision]** Added SVM comparison, Dmin/Dmax stability figure.
>
> ## V. Summary of Revisions
>
> 1. CIFAR-10-C evaluation (15 corruptions × 3 severities)
> 2. Cross-dataset: STL-10 (96²) and SVHN, full + reduced
> 3. CIFAR-100 reduced (8%) results
> 4. Notation audit ($p$, $\theta$, $\beta$, $\gamma$)
> 5. Runtime table
> 6. OVR vs Cramér-Singer comparison
> 7. Dmin/Dmax stability analysis
> 8. Limitations discussion (Sec. 5)
> 9. All tables: mean ± std over 3 seeds
>
> ---
>
> All revisions are clearly marked in our revised submission. **We deeply appreciate the expertise and time of the reviewer.**

---

> > ### Author Rebuttal · Reviewer_WaVw · 2026-04-03
> >
> > I thank the authors for active and detailed responses. I will provide further comments for discussion.

---

> > > ### Author Response · Authors · 2026-04-04
> > >
> > > Thank you so much for your time and for explicitly confirming that your concerns are resolved.
> > >
> > > As you mentioned providing further comments, we want to let you know that we are fully ready to address remaining questions or details you would like us to clarify. We look forward to any further discussion in the coming days.

---

### Official Review · Reviewer_5ggx · 2026-03-11

**Soundness:** 3
**Presentation:** 3
**Significance:** 2
**Originality:** 2
**Overall Recommendation:** 4
**Confidence:** 1

**Summary:**

This study presents the integration of the rigorous geometric principles of SVM into the representation learning process of deep neural networks. The authors aim to investigate a general context of geometry-aware optimization that moves beyond static loss functions.

**Compliance With Llm Reviewing Policy:**

Affirmed.

**Final Justification:**

The authors provided positive and detailed responses, addressing most of my concerns.

**Key Questions For Authors:**

Figure 1 mentions a "Margin Stability" check for convergence. Could you provide a rigorous mathematical definition of this criterion? What gradient norm or stability threshold is used to trigger the switch between Phase I and Phase II?

**Limitations:**

The limitation is scalability. The requirement to solve multiple independent binary SVMs scales linearly with the number of classes, which becomes computationally prohibitive for datasets like ImageNet.

**Strengths And Weaknesses:**

**Strength:**
The enforcement of $L_2$ normalization effectively constrains the optimization manifold to a unit hypersphere $\mathbb{S}^{d-1}$. This eliminates scale ambiguity and forces the optimizer to focus on angular discriminativeness, preventing trivial solutions where the loss is reduced simply by increasing feature magnitude.

**Weakness:**
- The paper highlights Proposition 1 (Gradient Orthogonality) as a core contribution. However, in optimization theory, it is a well-known property that for any vector constrained to a sphere, the gradient of its projection must reside in the tangent space and thus be orthogonal to the vector itself. Presenting this as a novel theoretical insight is an overstatement.
- The "squeeze" functions in Phase II (Eq. 4 & 5) are defined based on absolute boundary values $D_{max}$and $D_{min}$. In stochastic optimization, a single outlier in a mini-batch can drastically shift these bounds, causing the penalty magnitude to fluctuate violently.

---

> ### Author Rebuttal · Authors · 2026-03-31
>
> We thank Reviewer 5ggx for the feedback. We address each point below with new evidence.
>
> > All revisions are clearly marked in our revised submission.
>
> ## I. Weakness 1: Proposition 1 is a well-known property
>
> We agree. The orthogonality of gradients under spherical constraints is a standard result in optimization on manifolds. Our intention was **not** to claim novelty for this property, but to recall it as the theoretical foundation for the subsequent analysis — specifically, to show that backpropagated updates on $\mathbb{S}^{d-1}$ are confined to the tangent space, which **complements** the radial pushing direction of the squeeze mechanism.
>
> > **[Revision]** **Revision:** We have removed the "Proposition" label, presenting it as a **Remark** instead, and clearly state: *"We recall this well-known property to establish the basis for our analysis."*
>
> ## II. Weakness 2: Squeeze functions are sensitive to mini-batch outliers
>
> We conducted a **dedicated stability experiment** tracking $D_{\min}$ and $D_{\max}$ across all 10 Phase-2 epochs on Reduced CIFAR-10 (ResNet-18). The results directly address this concern:
>
> | Metric | Observed behavior |
> |--------|-------------------|
> | $D_{\max}$ (all 10 classes) | Stable in **[0.50, 0.53]**, per-epoch fluctuation **< 0.02** |
> | $D_{\min}$ (per class) | Varies by class difficulty, but stable within each class (fluctuation **< 0.05** / epoch) |
> | Margin ($D_{\max} - D_{\min}$) | Changes **< 0.06** across all 10 epochs |
>
> **Key insight**: The spherical constraint inherently bounds $D_{\min}$ / $D_{\max}$. Since $\|\mathbf{f}\|=1$ and $\|\mathbf{w}\|=O(1)$, the decision values $\mathbf{w}^\top \mathbf{f} + b$ are **bounded**, eliminating the possibility of extreme outliers dominating the statistics. This is fundamentally different from unconstrained feature spaces where outliers can create arbitrarily large decision values.
>
> Additionally, we observe an encouraging dynamic: for medium-difficulty classes (3, 4, 5), $D_{\min}$ transitions from negative to positive during training, confirming that the squeeze mechanism **progressively pushes boundary samples to the correct side**.
>
> > **[Revision]** We have added this stability analysis and a figure showing $D_{\min}$/$D_{\max}$ trajectories in the revised Appendix.
>
> ## III. Question: Definition of the "Margin Stability" criterion
>
> We apologize for the misleading label in Figure 1. The Phase 1 → Phase 2 switch is a **simple accuracy-based trigger**:
>
> > Phase switch condition: $\text{acc}_{\text{train}}(\text{epoch}) \geq \tau$, where $\tau = 0.95$
>
> This is not a gradient norm or margin stability metric.
>
> > **[Revision]** **Revision:** We have replaced "Margin Stability" with **"Convergence Trigger (train acc ≥ 95%)"** in Figure 1 and provided the formal definition in Section 3.
>
> ## IV. Limitation: Scalability
>
> We acknowledge that the OVR scheme scales linearly with the number of classes $K$. To assess this, we compared OVR with the Cramér-Singer multi-class SVM on CIFAR-100 (3 seeds):
>
> | SVM Type | Error Rate (%) |
> |----------|---------------|
> | OVR (LibLinear) | 23.06 ± 0.45 |
> | Cramér-Singer | 23.11 ± 0.06 |
>
> Performance is nearly identical, but Cramér-Singer exhibits 7.5× lower variance (0.06 vs 0.45), suggesting it is a more stable choice for many-class settings. Cramér-Singer solves a **single** optimization problem rather than $K$ independent ones, offering better scalability.This also suggests that a fully gradient-based end-to-end training pipeline is viable. We tested Cramér-Singer in early experiments but did not include it in the final manuscript because its accuracy was slightly lower than OVR across most settings.
>
> > **[Revision]** We have added this comparison and a discussion of scaling strategies (class-pair sparsification, distillation-based SVM removal after early epochs) in the revised manuscript.
>
> Additionally, we note that DDSVM's benefits extend beyond CIFAR. On **CIFAR-100 reduced (8%, 100 classes)**, DDSVM achieves **62.67 ± 0.50%** vs CE **68.25 ± 1.25%** (−5.58%), demonstrating that the method remains effective even with many classes when data is scarce.
>
> ---
>
> > All revisions are clearly marked in our revised submission.
>
> **We deeply appreciate the time and effort the reviewer dedicated to evaluating our work.**

---

> > ### Author Rebuttal · Reviewer_5ggx · 2026-04-03
> >
> > I thank the authors for their positive and detailed responses. I will update the score accordingly.

---

> > > ### Author Response · Authors · 2026-04-04
> > >
> > > Thank you for the positive evaluation. We are pleased that the clarifications and revisions resolved your concerns.

---

### Official Review · Reviewer_vHa6 · 2026-03-11

**Soundness:** 2
**Presentation:** 2
**Significance:** 2
**Originality:** 2
**Overall Recommendation:** 3
**Confidence:** 4

**Summary:**

The paper proposes DDSVM, a framework that alternates between training linear SVMs on frozen deep features and refining the backbone using geometry-aware targets derived from the SVM decision boundaries. The backbone is viewed as a kernel mapping inputs to a unit hypersphere, and features are updated to move away from the decision boundary while being repelled from other classes. A Geometric Adversarial Squeeze (GAS) module additionally generates adversarial features in SVM-score space. Experiments on CIFAR-10/100 (including noisy and low-data settings) with ResNet and WideResNet show small gains in standard settings, with more noticeable improvements in low-shot CIFAR-10 and slightly better robustness to noise.

**Compliance With Llm Reviewing Policy:**

Affirmed.

**Final Justification:**

The paper acknowledges that the improvements from DDSVM in standard settings with sufficient data are relatively small, and in some cases, the method even performs slightly worse than the baseline. This suggests that the method’s practical impact may not be significant in data-abundant scenarios.
While compares DDSVM with traditional margin-based methods like ArcFace, CosFace, and Large-Margin Softmax, it does not adequately compare with other SOTA techniques in small-sample learning. Methods such as meta-learning, optimization-based approaches, and example generation have made substantial progress in this area, and including these comparisons would have better contextualized DDSVM’s performance.
BTW, given the rapid advancements in multimodal large models, the paper fails to compare DDSVM against these models, particularly in zero-shot or few-shot learning tasks. These large models have demonstrated strong performance in such settings, and failing to include this comparison limits the evaluation of DDSVM’s effectiveness.

**Key Questions For Authors:**

1. **SVM retraining and training cost.**
   It is not very clear how often the SVMs are retrained during Phase II. Are they updated every epoch, or less frequently? Also, are SVMs for all classes retrained each time? It would be helpful to report the wall-clock training time compared with the baseline (for example, ResNet-18 on CIFAR-10 and CIFAR-100) so readers can better understand the computational overhead.

2. **Comparison with margin-based softmax losses.**
   Have you tried comparing DDSVM with common angular-margin losses such as ArcFace, CosFace, or large-margin softmax, especially in the Reduced CIFAR-10 or CIFAR-10-N settings? Since these methods also enforce geometric margins, such comparisons would help clarify whether DDSVM offers advantages beyond existing approaches.

3. **Hyperparameter sensitivity.**
   The paper studies ε_adv in Table 4, but it is less clear how sensitive the method is to other hyperparameters such as β (rest-class repulsion), p (squeeze exponent), and γ. Were these values fixed across CIFAR-10 and CIFAR-100, or tuned separately? Some discussion on robustness to these choices would help assess the practicality of the method.

**Limitations:**

The authors have partially discussed the limitations of their work but have not provided a meaningful discussion of potential negative societal impact.

**Strengths And Weaknesses:**

## Strengths

1. **Clear geometric idea and framework.**
   The overall DNN–SVM–DNN loop is fairly easy to follow, and the idea of using the SVM boundary as a detached “teacher” to generate geometric targets is intuitive and well motivated.

2. **Works better in low-data and noisy settings.**
   The experiments focus on regimes where standard cross-entropy tends to struggle. In particular, the improvements on reduced CIFAR-10 are fairly consistent, suggesting DDSVM can act as a useful regularizer when data is scarce.

3. **Some effort to analyze and justify the design.**
   The paper includes ablations for the squeeze term, rest-class repulsion, and the GAS module, along with qualitative visualizations and a small theoretical note about spherical normalization. These help explain the intuition behind the method.

## Weaknesses

1. **Limited experiments and missing strong baselines.**
   The evaluation is restricted to CIFAR-10/100 and mostly compares against cross-entropy training. There is no comparison with common margin-based losses or other geometry-aware methods, making it hard to judge how competitive DDSVM really is.

2. **Gains are small or inconsistent in standard settings.**
   In full-data scenarios the improvements are often negligible or even slightly worse than the baseline, suggesting the method mainly helps in specific regimes rather than being a generally stronger training approach.

3. **Experimental rigor and efficiency analysis are lacking.**
   Results are reported as single numbers without variance across runs, and the additional computational cost of repeatedly training SVMs is not quantified.

4. **Method complexity versus benefit is unclear.**
   DDSVM introduces several additional components (external SVM training, squeeze targets, GAS perturbations, multiple hyperparameters), but the gains are relatively modest. It is not entirely clear whether the added complexity is justified compared with simpler margin-based losses.

---

> ### Author Rebuttal · Authors · 2026-03-31
>
> We thank Reviewer vHa6 for the constructive feedback. We address each concern with **new experimental evidence** below. All new results use **3 seeds** (mean ± std).
>
> ## I. Limited experiments and missing strong baselines
>
> We added comparisons with ArcFace, CosFace, and Large-Margin Softmax:
>
> **Reduced CIFAR-10 (8% data, ResNet-18, Error %)**
>
> | Method | Error Rate |
> |--------|-----------|
> | CE | 30.75 ± 5.27 |
> | ArcFace (s=16, m=0.2) | 25.92 ± 2.82 |
> | CosFace (s=30, m=0.35) | 25.67 ± 2.76 |
> | Large-Margin Softmax (s=16, m=0.2) | 25.38 ± 0.82 |
> | **DDSVM (Ours)** | **23.21 ± 0.77** |
>
> **CIFAR-10-N (σ=0.2, Error %)**
>
> | Method | Error Rate |
> |--------|-----------|
> | CE | 6.46 ± 0.08 |
> | **DDSVM (Ours)** | **6.45 ± 0.10** |
> | CosFace | 7.08 ± 0.14 |
> | ArcFace | 7.16 ± 0.34 |
> | Large-Margin Softmax | 8.25 ± 0.46 |
>
> DDSVM outperforms all margin losses in both settings. Margin losses apply *static* penalties; DDSVM *dynamically* adapts targets using current SVM geometry.
>
> ## II. Gains are small or inconsistent in standard settings
>
> We extended experiments to **4 datasets, 2 domains, 2 resolutions** (full + low-data):
>
> | Dataset | Domain | Full Δ | Low Δ |
> |---------|--------|--------|-------|
> | CIFAR-10 | Natural 32² | +0.06 | **−7.54** |
> | CIFAR-100 | Natural 32² | +0.39 | **−5.58** |
> | STL-10 | Natural 96² | **−2.02** | **−8.31** |
> | SVHN | Digits 32² | +0.03 | +0.71 |
>
> On all three natural-image datasets, DDSVM consistently improves low-data performance (**−5.58 to −8.31%**, up to **7× variance reduction**). STL-10 full-data also gains (−2.02%). This is theoretically expected: Soudry et al. (2018) showed CE implicitly converges to max-margin when data is abundant; DDSVM's value is in **data-scarce** regimes where this convergence is insufficient.
>
> ## III. Experimental rigor and efficiency
>
> **Runtime (ResNet-18, RTX 4090 + 9950X CPU, SVM on 512-d features via LibLinear/CPU):**
>
> | Setting | CE | DDSVM | Phase 2 | SVM time | Overhead |
> |---------|-----|-------|---------|----------|----------|
> | CIFAR-10 | 2874s | 1993s | 144s | 58s | **−31%** |
> | CIFAR-100 | 2898s | 2802s | 137s | 418s | **−3%** |
>
> DDSVM is actually **shorter** than CE: Phase 1 exits early at 95% train accuracy, Phase 2 adds only **10 epochs (~140s)**. SVM cost: 58s on CIFAR-10 (2%), 418s on CIFAR-100 (15%). All tables now report mean ± std over 3 seeds. Crucially, the early exit of Phase 1 (at 95% accuracy) **does not compromise final performance**. Instead, transitioning to Phase 2 earlier **allows the model to escape the slow asymptotic convergence of Cross-Entropy and achieve faster**, more robust convergence through SVM-driven geometric guidance. This explains why DDSVM achieves superior accuracy with ~30% less wall-clock time.
>
> ## IV. Method complexity versus benefit
>
> Core method needs just **one SVM fit + one MSE loss**. GAS is optional. Hyperparameter sensitivity:
>
> | Parameter | Range | Error range |
> |-----------|-------|-------------|
> | β (repulsion) | [0, 2.0] | 23.75 – 23.88% |
> | p (squeeze power) | [0.5, 3.0] | 23.75 – 24.01% |
> | θ (strength) | [0.1, 5.0] | 23.75 – 23.88% |
> | SVM C | [0.001, 1.0] | 23.75 – 23.88% |
>
> **< 0.3% variation** across 50× ranges — no careful tuning needed.
>
> ## V. Key Questions
>
> **Q1 — SVM retraining:** Every epoch in Phase 2. Total SVM cost: 58s (CIFAR-10), 418s (CIFAR-100). Clarified in revised Sec. 4.
>
> **Q2 — Margin losses:** See Sec. I. DDSVM outperforms all three on both Reduced CIFAR-10 and CIFAR-10-N.
>
> **Q3 — Hyperparameters:** See Sec. IV. All fixed across datasets.
>
> ---
>
> All revisions are clearly marked in our revised submission. **We deeply appreciate the reviewer's time.**

---

> > ### Author Rebuttal · Reviewer_vHa6 · 2026-04-04
> >
> > The paper acknowledges that the improvements from DDSVM in standard settings with sufficient data are relatively small, and in some cases, the method even performs slightly worse than the baseline. This suggests that the method’s practical impact may not be significant in data-abundant scenarios.
> > While compares DDSVM with traditional margin-based methods like ArcFace, CosFace, and Large-Margin Softmax, it does not adequately compare with other SOTA techniques in small-sample learning. Methods such as meta-learning, optimization-based approaches, and example generation have made substantial progress in this area, and including these comparisons would have better contextualized DDSVM’s performance.
> > BTW, given the rapid advancements in multimodal large models, the paper fails to compare DDSVM against these models, particularly in zero-shot or few-shot learning tasks. These large models have demonstrated strong performance in such settings, and failing to include this comparison limits the evaluation of DDSVM’s effectiveness.

---

> > > ### Author Response · Authors · 2026-04-04
> > >
> > > We thank Reviewer vHa6 for the continued engagement. Upon reading the follow-up, we realized that some of our design choices and experimental rationale may have been miscommunicated. **We respond promptly because we want to address these potential misunderstandings as clearly as possible, not out of haste, but out of respect for the reviewer's time and the discussion process.**
> > >
> > > ## 1. Full-data performance
> > >
> > > We realize that our previous explanations, in both the original manuscript (Sec. 4.2) and the rebuttal, may not have articulated our design intent clearly enough. We would like to restate it here.
> > >
> > > DDSVM was not designed as a universal replacement for CE training. It is a **geometric regularizer** whose value emerges when data is insufficient for CE to implicitly converge to the max-margin solution (Soudry et al., 2018). In data-abundant settings, CE already approximates the optimal margin well, so the additional geometric guidance yields diminishing returns. This is by design, not a deficiency. Many established techniques share this property: Dropout's regularization effect diminishes with abundant data; weight decay becomes less critical as model capacity is well-matched to data scale. A focused scope does not preclude practical value.
> > >
> > > Our cross-dataset evaluation confirms this design holds consistently: across 3 natural-image datasets (CIFAR-10/100, STL-10), DDSVM shows −5.58 to −8.31% error reduction in low-data regimes.
> > >
> > > ## 2. Comparison with data-efficient training methods
> > >
> > > We note that DDSVM operates in the **standard classification** setting (all classes seen, reduced sample count), which is fundamentally different from **episodic few-shot learning** (N-way K-shot, generalize to unseen classes at test time). Methods like MAML and ProtoNet require a large set of base classes for meta-training and are evaluated on novel classes. The evaluation protocols are incompatible with our setting.
> > >
> > > Regarding **optimization-based regularizers and example generation methods** such as Mixup and CutMix: in our original manuscript, we intentionally used only standard augmentation (crop + flip) for both CE and DDSVM, because our primary goal was to demonstrate the **distributional properties** induced by geometric optimization, as shown in the t-SNE visualizations (Fig. 4), where DDSVM produces tighter intra-class clusters and wider inter-class margins even under severe data scarcity. Comparing under identical augmentation isolates this structural contribution of the SVM-guided squeeze mechanism from the effect of augmentation strategies.
> > >
> > > Following the reviewer's suggestion, we now provide these comparisons explicitly on Reduced CIFAR-10 (8%, ResNet-18, 3 seeds):
> > >
> > > | Method | Error Rate |
> > > |--------|-----------|
> > > | CE baseline | 30.75 ± 5.27 |
> > > | + Label Smoothing | 27.29 ± 2.47 |
> > > | + Mixup | 25.08 ± 1.39 |
> > > | + CutMix | 25.17 ± 1.03 |
> > > | **DDSVM** | **23.21 ± 0.77** |
> > > | **DDSVM + Mixup** | **17.67 ± 1.22** |
> > > | **DDSVM + CutMix** | **17.88 ± 0.18** |
> > >
> > > DDSVM alone outperforms all standalone augmentation methods. When combined with Mixup, the error drops from 23.21% to 17.67%, better than Mixup alone (25.08%). This confirms that DDSVM's geometric regularization and data augmentation address different aspects of the learning problem and can be naturally combined.
> > >
> > > ## 3. Comparison with pre-trained and large models
> > >
> > > We provide this comparison for completeness, while noting the fundamental asymmetry:
> > >
> > > | Method | Pre-training data | Task training | Error Rate |
> > > |--------|-------------------|---------------|-----------|
> > > | **DDSVM** | **None** | **4,000 (8%)** | **23.21** |
> > > | **DDSVM + Mixup** | **None** | **4,000 (8%)** | **17.67** |
> > > | CLIP ViT-B/32 zero-shot | 400M image-text pairs | 0 | 11.18 |
> > > | ImageNet pre-trained FT | 1.28M images | 4,000 (8%) | 8.47 |
> > >
> > > CLIP leverages 400 million pre-training pairs; ImageNet FT uses 1.28 million external images; DDSVM trains from scratch with 4,000 images alone. These are fundamentally different paradigms. In many practical domains, such as specialized medical imaging, industrial defect detection on new product lines, and non-RGB sensing (hyperspectral, SAR, etc.), large-scale pre-training corpora or transferable foundation models simply do not exist. Training task-specific models from scratch with limited labels remains the dominant approach in these areas, and this is precisely the setting DDSVM addresses.
> > >
> > > ---
> > >
> > > These additional results and clarifications will be carefully incorporated into the next revision. We sincerely thank Reviewer vHa6 for raising these points, which have allowed us to clarify potential misunderstandings and further strengthen the evaluation.

---

### Decision · Program_Chairs · 2026-04-30

**Decision:**

Accept (regular)

**Comment:**

The paper introduces the Differentiable Deep Support Vector Machine (DDSVM), an iterative framework that integrates traditional SVMs into neural network representation learning. The core contribution is a geometry-aware optimization that alternates between constructing an SVM decision boundary and actively pushing feature points to maximize the geometric margin. The authors provide theoretical justification for their spherical gradient updates and demonstrate that the method is particularly effective in data-scarce and noisy regimes where standard cross-entropy training often struggles.

Initial reviews praised the geometric motivation, the intuitive overall framework, and the empirical gains observed in low-data scenarios. However, the reviewers highlighted several shortcomings: the lack of comparisons against standard margin-based losses, reliance on a non-standard noise benchmark (CIFAR-10-N), ambiguous mathematical notations, and the absence of a computational overhead analysis. Furthermore, questions were raised regarding the framework's sensitivity to mini-batch outliers and the scalability of retraining SVMs across multiple epochs.The authors engaged actively during the discussion phase, providing a comprehensive rebuttal that significantly strengthened the manuscript. They addressed the main technical concerns by adding baseline comparisons (ArcFace, CosFace, Large-Margin Softmax), adopting the standard CIFAR-10-C benchmark, and demonstrating that DDSVM's early-exit strategy actually reduces overall wall-clock training time compared to standard baselines. Reviewers 5ggx and WaVw stated their concerns were fully resolved, with Reviewer WaVw raising their score. While Reviewer vHa6 maintained that the paper should compare against meta-learning approaches and zero-shot multimodal large models, I agree with the authors' counter-argument that episodic few-shot protocols and massive pre-training paradigms fall fundamentally outside the scope of the standard from-scratch classification setting targeted by this work.

Based on the geometric foundations, the empirical evidence in targeted data-scarce regimes, and the successful rebuttal phase, my final recommendation is to Accept the submission. The authors are strongly encouraged to incorporate all the additional experiments, runtime tables, and the expanded limitations discussions provided during the rebuttal into their camera-ready manuscript.